# BAYESIAN EXPLORATION NETWORKS

## ABSTRACT

Bayesian reinforcement learning (RL) offers a principled and elegant approach for sequential decision making under uncertainty. Most notably, Bayesian agents do not face an exploration/exploitation dilemma, a major pathology of frequentist methods. A key challenge for Bayesian RL is the computational complexity of learning Bayes-optimal policies, which is only tractable in toy domains. In this paper we propose a novel model-free approach to address this challenge. Rather than modelling uncertainty in high-dimensional state transition distributions as model-based approaches do, we model uncertainty in a one-dimensional Bellman operator. Our theoretical analysis reveals that existing model-free approaches either do not propagate epistemic uncertainty through the MDP or optimise over a set of contextual policies instead of all history-conditioned policies. Both approximations yield policies that can be arbitrarily Bayes-suboptimal. To overcome these issues, we introduce the Bayesian exploration network (BEN) which uses normalising flows to model both the aleatoric uncertainty (via density estimation) and epistemic uncertainty (via variational inference) in the Bellman operator. In the limit of complete optimisation, BEN learns true Bayes-optimal policies, but like in variational expectation-maximisation, partial optimisation renders our approach tractable. Empirical results demonstrate that BEN can learn true Bayes-optimal policies in tasks where existing model-free approaches fail.

## 1 INTRODUCTION

*Detailed proofs for theorems and examples are provided in Appendix C*

In reinforcement learning (RL), an agent is tasked with learning an optimal policy that maximises expected return in a Markov decision process (MDP). In most cases, the agent is in a learning setting and does not know the underlying MDP a priori: typically the reward and transition distributions are unknown. A Bayesian approach to reinforcement learning characterises the uncertainty in unknown governing variables in the MDP by inferring a posterior over their values conditioned on observed histories of interactions. Using the posterior it is possible to marginalise across unknown variables and derive a belief transition distribution that characterises how the uncertainty will evolve over all future timesteps. The resulting Bayesian RL (BRL) objective transforms a learning problem into a planning problem with a well defined set of optimal policies, known as Bayes-optimal policies which are a gold standard for exploration (Martin, 1967; Duff, 2002). From this perspective, the exploration/exploitation dilemma is a major pathology of frequentist RL due to the violation of the *conditionality principle*: when in a learning problem, frequentist methods can condition on information that the agent does not have access to, namely the unknown transition and reward distributions. Frequentist RL researchers must close this gap by developing exploration heuristics as there is no formal method to tackling this dilemma. In contrast Bayes-optimal policies solve the exploration/exploitation dilemma by exploring to reduce epistemic uncertainty in the MDP, but only insofar as that reduction in uncertainty increases expected returns as the belief evolves across timesteps. Moreover any non-Bayesian policy is suboptimal in terms of optimising the expected returns according to the belief induced by the prior and model of the state and reward transition distributions.

Despite the formal theoretical benefits, learning Bayes-optimal policies that scale to domains beyond toy examples remains a significant challenge due to several sources of intractability. Firstly, model-based approaches must maintain a posterior over a model of the state transition dynamics, which is notoriously computationally complex for even low dimensional state spaces (Wasserman, 2006). Secondly, even if it is tractable to calculate and maintain the posterior, the marginalisation needed to find the Bayesian transition and reward distributions requires high dimensional integrals. Finally,

given the Bayesian distributions, a planning problem must then be solved in belief space for every history-augmented state to obtain the Bayes-optimal policy.

Alternatively, model-free approaches characterise uncertainty in a Bellman operator. This avoids the issues of modelling uncertainty in high dimensional transition distributions, as Bellman operators require the specification of a one-dimensional conditional distribution. Whilst existing model-free approaches to BRL exist, our key contribution is to provide a rigorous theoretical analysis demonstrating that all existing methods inadvertently solve a myopic or contextual approximation to the true Bayesian objective, which prevents them from learning a true Bayes-optimal policy. Our novel formulation rewrites the Bayesian Bellman operator as an expectation over optimal Bellman operators using the posterior over MDPs. This allows uncertainty to be characterised in optimal Bellman operators, whilst still solving the true BRL objective, with the corresponding optimal policy being Bayes-optimal.

Motivated by these shortcomings, we introduce a Bayesian exploration network (BEN) for model-free BRL that is exactly equivalent to modelling uncertainty in a transition and reward function using a model-based approach. BEN first reduces the dimensionality of inputs to a one-dimensional variable using a $Q$-function approximator. The output is then passed through a Bayesian network, which significantly reduces the dimensionality of Bayesian parameters we must infer a posterior over. The $Q$-function approximator parameters can then be found by solving a Bayesian Bellman equation. Moreover, like in an actor-critic approach, BEN can be trained using partial stochastic gradient descent methods at each timestep, bypassing computational complexity issues associated with finding a Bayes-optimal policy. This comes at the expense of learning an approximately Bayes-optimal policy instead but one that converges to the true Bayes-optimal policy in the limit of complete optimisation.

To verify our theoretical claims, we evaluate BEN in a search and rescue environment, which is a novel higher dimensional variant of the tiger problem (Kaelbling et al., 1998). We show BEN solves the task while oracles of existing state-of-the-art model-free BRL approaches based on BootDQN+Prior (Osband et al., 2018) and Bayesian Bellman Actor Critic (Fellows et al., 2021) fail due to their inability to learn Bayes-optimal policies. Moreover, our results show that whilst in the limit of complete optimisation, BEN recovers true Bayes-optimal policies, complete optimisation is not necessary as BEN behaves near Bayes-optimally after taking a small number of optimisation steps on our objective for every observation.

## 2  BAYESIAN AND FREQUENTIST REINFORCEMENT LEARNING

### 2.1  FREQUENTIST RL

We define a space of infinite-horizon, discounted Markov decision processes (MDPs) by introducing a variable $\phi \in \Phi \subseteq \mathbb{R}^d$: $\mathcal{M}(\phi) := \langle \mathcal{S}, \mathcal{A}, P_0, P_S(s, a, \phi), P_R(s, a, \phi), \gamma \rangle$ where each $\phi$ indexes a specific MDP by parametrising a transition distribution $P_S(s, a, \phi) : \mathcal{S} \times \mathcal{A} \times \Phi \to \mathcal{P}(\mathcal{S})$ and reward distribution $P_R(s, a, \phi) : \mathcal{S} \times \mathcal{A} \times \Phi \to \mathcal{P}(\mathbb{R})$. We denote the corresponding joint conditional state-reward distribution as $P_{R,S}(s, a, \phi)$. We assume that the agent has complete knowledge of the set of states $\mathcal{S}$, set of actions $\mathcal{A}$, initial state distribution $P_0 \in \mathcal{P}(\mathcal{S})$ and discount factor $\gamma$. A frequentist agent follows a policy $\pi : \mathcal{S} \times \Phi \to \mathcal{P}(\mathcal{A})$, taking actions $a_t \sim \pi(s_t, \phi)$. We denote the set of all MDP conditioned policies as $\Pi_\Phi := \{\pi : \mathcal{S} \times \Phi \to \mathcal{P}(\mathcal{A})\}$. Given an initial state $s_0 \sim P_0$ we denote a trajectory of future state-action-rewards up to state $s_t$ at time $t$ as the sequence: $\tau_t := \{s_0, a_0, r_0, s_1, a_1, r_1, \ldots a_{t-1}, r_{t-1}, s_t\} \in \mathcal{T}_t$ where $\mathcal{T}_t := \mathcal{S} \times \mathcal{A} \times \mathbb{R} \ldots \mathcal{A} \times \mathbb{R} \times \mathcal{S}$ is the corresponding product space. We denote the distribution over trajectory $\tau_t$ as: $P_t^\pi(\phi)$.

In the infinite horizon, discounted setting, the goal of a frequentist agent is to find a policy that optimises the objective: $J^\pi(\phi) = \mathbb{E}_{\tau_\infty \sim P_\infty^\pi(\phi)} [\sum_{t=0}^\infty \gamma^t r_t]$. We denote an optimal policy as: $\pi^\star(\cdot, \phi) \in \Pi_\Phi^\star(\phi) := \arg \sup_{\pi \in \Pi_\Phi} J^\pi(\phi)$, where $\Pi_\Phi^\star(\phi)$ is the set of all optimal MDP-conditioned policies that are optimal for $\mathcal{M}(\phi)$. For an optimal policy $\pi^\star$, the optimal quality function ($Q$-function) $Q^\star : \mathcal{S} \times \mathcal{A} \times \Phi \to \mathbb{R}$ satisfies the optimal Bellman equation: $\mathcal{B}^\star [Q^\star] (s_t, a_t, \phi) = Q^\star(s_t, a_t, \phi)$ where $\mathcal{B}^\star [Q^\star] (s_t, a_t, \phi) := \mathbb{E}_{r_t, s_{t+1} \sim P_{R,S}(s_t, a_t, \phi)}[r_t + \sup_{a' \in \mathcal{A}} Q^\star(s_{t+1}, a', \phi)]$ is the optimal Bellman operator.

If the agent has access to the true MDP $\mathcal{M}(\phi^\star)$, computational complexity issues aside, an optimal policy can be obtained by solving a *planning* problem, either by optimising the RL objective $J^\pi(\phi^\star)$ directly for $\pi$ or by solving an optimal Bellman equation and taking the action $a_t \in \arg \sup_{a' \in \mathcal{A}} Q^\star(s_t, a', \phi^\star)$. In the more realistic setting, the agent does not have access to the

MDP's transition dynamics and/or reward function, and must balance learning these variables through exploration of the MDP at the cost of behaving suboptimally with solving the underlying planning problem by exploiting the information it has observed. This setting is known as a *learning* problem and solving the exploration/exploitation dilemma remains a major challenge for any agent learning to behave optimally.

## 2.2  BAYESIAN RL

A Bayesian epistemology characterises the agent's uncertainty in the MDP through distributions over $\Phi$. We start by defining the prior distribution $P_\Phi$ which represents the a priori belief in the true value $\phi^\star$ before the agent has observed any transitions. Priors are a powerful aspect of BRL, allowing us to provide the agent with any information about the MDP and transfer knowledge between agents and domains. In the tabula rasa setting, priors can be uninformative; can be used to encode optimism or pessimism in unknown states; or a minimax prior representing the worst possible prior distribution over MDPs an agent could face (Buening et al., 2023). As the agent interacts with the environment, it observes a history of data $h_t \coloneqq \{s_0, a_0, r_0, s_1, a_1, r_1, \ldots a_{t-1}, r_{t-1}, s_t\} \in \mathcal{H}_t$ where $\mathcal{H}_t$ is the corresponding state-action-reward product space. Given a set of historical data $h_t$, we aim to reason over future trajectories; thus Bayesian agents follow policies that condition on histories, rather than single states. We denote the space of all trajectories $\mathcal{H} \coloneqq \{\mathcal{H}_t | t \geq 0\}$ and the set of all trajectory conditioned policies as $\Pi_\mathcal{H} \coloneqq \{\pi : \mathcal{H} \to \mathcal{P}(\mathcal{A})\}$. A Bayesian agent characterises the uncertainty in the MDP by inferring the posterior $P_\Phi(h_t)$ for each $t \geq 0$.

The prior is a special case of the posterior with $h_t = \varnothing$. The posterior $P_\Phi(h_t)$ represents the agent's beliefs in the MDP and can be used to *marginalise* across all MDPs according the agent's uncertainty. This yields the Bayesian state-reward transition distribution: $p_{R,S}(h_t, a_t) \coloneqq \mathbb{E}_{\phi \sim P_\Phi(h_t)}[P_{R,S}(s_t, a_t, \phi)]$. Given this distribution we can reason over counterfactual future trajectories $\tau_{t:t'} \coloneqq \{a_t, r_t, s_{t+1}, \ldots s_{t'}\}$ using the predictive distribution over trajectories conditioned on $h_t$, which we denote as $P_{t:t'}^\pi(h_t)$ with density: $P_{t:t'}^\pi(\tau_{t:t'}|h_t) = \prod_{i=t}^{t'-1} \pi(a_i|h_i)p(r_i, s_{i+1}|h_i, a_i)$. Using the predictive distribution, we define the BRL objective as: $J_{\text{Bayes}}^\pi \coloneqq \mathbb{E}_{\tau_{0:\infty} \sim P_{0:\infty}^\pi}\left[\sum_{i=0}^\infty \gamma^i r_i\right]$. A corresponding optimal policy is known as a Bayes-optimal policy, which we denote as: $\pi_{\text{Bayes}}^\star(\cdot) \in \Pi_{\text{Bayes}}^\star \coloneqq \arg\sup_{\pi \in \Pi_\mathcal{H}} J_{\text{Bayes}}^\pi$.

Unlike in frequentist RL, Bayesian variables depend on histories obtained through posterior marginalisation; hence the posterior is often known as the *belief state*, which augments each ground state $s_t$ like in a partially observable MDP. Analogously to the state-transition distribution in frequentist RL, we can define a belief transition distribution $P_\mathcal{H}(h_t, a_t)$ using the Bayesian reward and transition distributions, which has the density $p_\mathcal{H}(h_{t+1}|h_t, a_t) = p(s_{t+1}, r_t|h_t, a_t) \underbrace{p(h_t, a_t|h_t, a_t)}_{=1} = p(s_{t+1}, r_t|h_t, a_t)$. Using the belief transition, we define the *Bayes-adaptive MDP* (BAMDP) (Duff, 2002): $\mathcal{M}_{\text{BAMDP}} \coloneqq \langle \mathcal{H}, \mathcal{A}, P_0, P_\mathcal{H}(h, a), \gamma \rangle$, which can be solved using planning methods to obtain a Bayes-optimal policy (Martin, 1967).

A Bayes-optimal policy naturally balances exploration with exploitation: after every timestep the agent's uncertainty is characterised via the posterior conditioned on the history $h_t$, which includes all future trajectories to marginalise over. The BRL objective therefore accounts for how the posterior evolves after each transition, and hence any Bayes-optimal policy $\pi_{\text{Bayes}}^\star$ is optimal not only according to the epistemic uncertainty at a single timestep, but also to the epistemic uncertainty at every future timestep, decaying at a rate according to the discount factor.

Unlike in frequentist RL, if the agent is in a learning problem, finding a Bayes-optimal policy is always possible given sufficient computational resources. This is because any uncertainty in the MDP is marginalised over according to the belief characterised by the posterior. BRL thus does not suffer from the exploration/exploitation dilemma as actions are sampled from optimal policies that only condition on historical observations $h_t$, rather than the unknown MDP $\phi^\star$. More formally, this is a direct consequence of the *conditionality principle*, which all Bayesian methods adhere to, meaning that Bayesian decisions never condition on data that the agent has not observed. From this perspective, the exploration/exploitation dilemma is a pathology that arises because frequentist approaches violate the conditionality principle.

For a Bayes-optimal policy $\pi^\star$, we define the optimal Bayesian $Q$-function as $Q^\star(h_t, a_t) \coloneqq Q^{\pi_{\text{Bayes}}^\star}(h_t, a_t)$, which satisfies the optimal Bayesian Bellman equation $Q^\star(h_t, a_t) = \mathcal{B}^\star[Q^\star](h_t, a_t)$

where:

$$\mathcal{B}^{\star}[Q^{\star}](h_t, a_t) = \mathbb{E}_{h_{t+1} \sim P_{\mathcal{H}}(h_t, a_t)} \left[ r_t + \gamma \sup_{a'} Q^{\star}(h_{t+1}, a') \right], \tag{1}$$

is the optimal Bayesian Bellman operator. It is possible to construct a Bayes-optimal policy by choosing the action that maximises the optimal Bayesian $Q$-function $a_t \in \arg\sup_{a'} Q^{\star}(h_t, a')$; hence learning $Q^{\star}(h_t, \cdot)$ is sufficient for solving the BAMDP. We take this value based approach in this paper.

## 3 RELATED WORK

BEN is the first model-free approach to BRL that can learn Bayes-optimal policies. To relate BEN to other approaches, we clarify the distinction between model-free and model-based BRL:

**Definition 1.** *Model-based approaches define a prior $P_\Phi$ over and a model of the MDP's state and reward transition distributions: $P_S(s, a, \phi)$ and $P_R(s, a, \phi)$. Model-free approaches define a prior $P_\Phi$ over and a model of the MDP's Bellman operators (or Q-functions): $P_B(\cdot, \phi)$ (or $P_Q(\cdot, \phi)$).*

This definition mirrors classical interpretations of model-based and model-free RL, which categorises algorithms according to whether a model of transition dynamics is learnt or the Q-function is estimated directly (Sutton and Barto, 2018). We prove in Theorem 3 that due to *the sufficiency principle*, whichever approach is taken, a Bayes-optimal policy may still be learnt, and is key to proving Bayes-optimality of BEN. A further detailed discussion of this core contribution can be found in Appendix C.1.

As many real-world problems of interest have high dimensional state spaces, representing the transition distribution accurately requires a model $P_S(s, a, \phi)$ with a large number of parameters. This further compounds the intractability issues associated with taking a model-based approach to solving the BAMDP as a posterior needs to be inferred over an infeasible number of parameters and marginalisation involves higher dimensional integrals. Moreover, the sample efficiency for density estimation of conditional distributions scales poorly with increasing dimensionality (Grünewälder et al., 2012): Wasserman (2006) show that when using a nonparametric frequentist kernel approach to density estimation, even with an optimal $4 + d$ bandwidth, the mean squared error scales as $\mathcal{O}(N^{\frac{-4}{d+4}})$ where $N$ is the number of samples from the true density - to ensure a mean squared error of less than 0.1 when the target density is a multivariate Gaussian of dimension 10, the number of samples required is 842000 in comparison to 19 for a 2 dimensional problem. In a parametric approach, this implies that the number of parameters required to sufficiently represent more realistic multi-modal transition distributions will scale poorly with increasing dimension of the state-space. From a Bayesian perspective, we would expect the posterior to concentrate at a slower rate with increasing dimensionality as the agent would require more data to decrease its uncertainty in the transition model parameters. We provide a review of several model-based approaches and their approximations in Appendix B

In contrast, the majority of existing model-free approaches attempt to infer a posterior over $Q$-functions $P_Q^\pi(h_t)$ given a history of samples $h_t$, thus requiring a model of the aleatoric uncertainty in the $Q$-function samples $q \sim P_Q^\pi(s, a, \phi)$. $P_Q^\pi(s, a, \phi) : \mathcal{S} \times \mathcal{A} \times \Phi \to \mathcal{P}(\mathbb{R})$ is typically a parametric Gaussian, which is a conditional distribution over a one-dimensional space, allowing for standard techniques from Bayesian regression to be applied. As inferring a posterior over $Q$-functions requires samples from complete returns, some degree of bootstrapping using function approximation is required for algorithms to be practical (Gal and Ghahramani, 2016; Osband et al., 2018; Fortunato et al., 2018; Lipton et al., 2018; Osband et al., 2019; Touati et al., 2019). By introducing bootstrapping, model-free approaches actually infer a posterior over *Bellman operators*, which concentrates on the true Bellman operator with increasing samples under appropriate regularity assumptions (Fellows et al., 2021). Instead of attempting to solve the BAMDP exactly, existing model-free approaches employ posterior sampling where a single MDP is drawn from the posterior at the start of each episode (Thomson, 1933; Strens, 2000; Osband et al., 2013), or optimism in the face of uncertainty (OFU) (Lai and Robbins, 1985; Kearns and Singh, 2002) where exploration is increased or decreased by a heuristic to reflect the uncertainty characterised by the posterior variance (Jin et al., 2018; Ciosek et al., 2019; Luis et al., 2023). Unfortunately, both posterior sampling and OFU exploration can be highly inefficient and far from Bayes-optimal (Zintgraf et al., 2020; Buening et al., 2023). Exploration strategies aside, a deeper issue with existing model-free Bayesian approaches is that an optimal policy under their formulations is not Bayes-optimal, but instead solves either a *myopic* or *contextual* approximation to the BRL objective. We now investigate this problem further in Section 4.

# 4 SHORTCOMINGS OF MODEL-FREE APPROACHES

As motivated in Section 3, modelling uncertainty in a low-dimensional variable such as a value function or Bellman operator is clearly desirable; however naively defining a model over any variable in the MDP may result in policies that are not Bayes-optimal. We now take a theoretical look at the approximations implicit in existing model-free approaches to BRL to recover the objectives that are actually optimised.

## 4.1 CONTEXTUAL BRL

If we make the simplifying assumption that the set of Bayesian policies can be represented by the set of MDP-conditioned policies $\Pi_\Phi$ using the posterior to marginalise over $\phi$, we can define a set of *contextual* policies: $\Pi_{\text{Contextual}} := \left\{ \mathbb{E}_{\phi \sim P_\Phi(\mathcal{H}_t)} \left[ \pi(\cdot, \theta) \right] | \pi \in \Pi_\Phi \right\}$. Clearly $\Pi_{\text{Contextual}} \subset \Pi_{\mathcal{H}}$ but it is not obvious whether it is possible to obtain a Bayes-optimal policy using the set of contextual policies in place of the full set of Bayesian policies $\Pi^\star_{\text{Bayes}}$. To answer this question, we first define the set of optimal contextual policies as: $\Pi^\star_{\text{Contextual}} := \arg \sup_{\pi \in \Pi_{\text{Contextual}}} J^\pi_{\text{Bayes}}$, which we relate to the set of optimal MDP-conditioned policies using the following theorem:

**Theorem 1.** *Contextual Bayesian value functions and optimal policies can be related to frequentist value functions and optimal policies through marginalisation, that is:* $\Pi^\star_{\text{Contextual}} = \left\{ \mathbb{E}_{\phi \sim P_\Phi(h_t)} \left[ \pi^\star(\cdot, \phi) \right] | \pi^\star(\cdot, \phi) \in \Pi^\star_\Phi \right\}, Q^\star_{\text{Contextual}}(h_t, a) = \mathbb{E}_{\phi \sim P(\phi|h_t)} \left[ Q^\star(s_t, a, \phi) \right].$

Theorem 1 proves that the set of contextual optimal policies $\Pi^\star_{\text{Contextual}}$ can only be formed from a mixture of optimal policies conditioned on specific MDPs using the posterior. We confirm this implies contextual optimal policies can be arbitrarily Bayes-suboptimal in Corollary 1.1, using the tiger problem (Kaelbling et al., 1998) as a counterexample:

**Corollary 1.1.** *There exist MDPs with priors such that* $\Pi^\star_{Contextual} \cap \Pi^\star_{Bayes} = \varnothing.$

Theorem 1 also proves that modelling uncertainty in an (optimal) $Q$-function is equivalent to learning a Bayesian (optimal) $Q$-function over the set of contextual policies: $\mathbb{E}_{q \sim P_Q(h_t, a)}[q] = \mathbb{E}_{\phi \sim P_\Phi(h_t)} \left[ \mathbb{E}_{q \sim P_Q(s_t, a, \phi)} [q] \right] = \mathbb{E}_{\phi \sim P_\Phi(h_t)} \left[ Q^\star(s_t, a, \phi) \right] = Q^\star_{\text{Contextual}}(h_t, a)$. Hence at best existing model-free approaches yield contextual optimal policies.

## 4.2 MYOPIC BRL

A further approximation to exact BRL, whether intentional or not, is to solve a *myopic* variation of the true BAMDP through methods such as a QMDP (Kaelbling et al., 1998). Here the distribution: $P_{R,S}(h_t, s_{t'}, a_{t'}) = \int P_{R,S}(s_{t'}, a_{t'}, \phi) dP_\Phi(\phi|h_t)$ is used to characterise the epistemic uncertainty over all future timesteps $t' \geq t$ and does not account for how the posterior evolves after each transition. The corresponding myopic distribution over a trajectory $\tau_{t:t'}$ is: $p^\pi_{\text{Myopic}}(\tau_{t:t'}|h_t) = \prod_{i=t}^{t'-1} \pi(a_i|s_i, h_t) \cdot p_{R,S}(r_i, s_{i+1}|s_i, a_i, h_t)$. Here, only $P_\Phi(h_t)$ is used to marginalise over uncertainty at each timestep $t' \geq t$ and information in $\tau_{t:t'}$ is not used to update the posterior.

Several existing model-free approaches (Gal and Ghahramani, 2016; Fortunato et al., 2018; Lipton et al., 2018; Touati et al., 2019) naively introduce a $Q$-function approximator $Q_\omega : \mathcal{S} \times \mathcal{A} \to \mathbb{R}$ whose parameters minimise the mean-squared Bayesian Bellman error: $\omega^\star \in \arg \min_{\omega \in \Omega} \|Q_\omega(s, a) - \mathcal{B}^\star_{\text{Myopic}}[Q_\omega](h_t, s, a)\|^2_\rho$ where $\mathcal{B}^\star_{\text{Myopic}}[Q_\omega]$ is the myopic Bellman operator:

$$\mathcal{B}^\star_{\text{Myopic}}[Q_\omega](h_t, s_{t'}, a_{t'}) = \mathbb{E}_{r_{t'}, s_{t'+1} \sim P_{R,S}(h_t, s_{t'}, a_{t'})} \left[ r_{t'} + \gamma \sup_{a'} Q_\omega(s_{t'+1}, a') \right], \quad (2)$$

thereby finding a myopic optimal policy instead of a true Bayes-optimal policy. Two notable exceptions are BootDQN+Prior (Osband et al., 2018; 2019) and its actor-critic analogue BBAC (Fellows et al., 2021); however these two approaches still only solve the BRL objective for contextual policies.

## 4.3 ALEATORIC UNCERTAINTY MATTERS

Accurately representing the aleatoric uncertainty through the model $P^\star_\omega(h_t, a, \phi)$ is the focus of distributional RL (Bellemare et al., 2017) and has been ignored by the model-free BRL community. As discussed in Section 3, most existing parametric model-free BRL approaches have focused on representing the epistemic uncertainty in the posterior under a parametric Gaussian model (Osband et al., 2018). One exception is Model-based $Q$-variance estimation (Luis et al., 2023); however this approach still optimises over contextual policies and relies on optimistic exploration bonuses, which like posterior sampling, are Bayes-suboptimal. To motivate the need to develop models with

improved capacity for modelling aleatoric uncertainty, we investigate the relationship between a model specified over MDPs in model-based BRL and the equivalent model-free distribution over Bellman operators in the following example:

**Example 1.** *Consider the space of MDPs with $\mathcal{S} = \mathbb{R}$, $P_{\mathcal{S}}(s_t, a_t, \phi) = \mathcal{N}(\mu_\phi(s_t, a_t), \sigma_\phi(s_t, a_t))$ and a deterministic reward $r_t = r(s_t, a_t)$ which is known a priori. For any $Q$-function approximator $Q_\omega(s, a)$ such that $v_t = V_\omega(s_t) := \sup_{a'} Q_\omega(s_t, a')$ with inverse $s_t = V_\omega^{-1}(v_t)$, the distribution over optimal Bellman operators under the transformation $b_t = r(s_t, a_t) + \gamma \sup_{a'} Q_\omega(s_t, a')$ has*

*density:* $p_B(b_t|s_t, a_t, \phi) = \left( \frac{|\partial_{v_t} V^{-1}(v_t)|}{\sqrt{2\pi\sigma_\phi(s_t, a_t)^2}} \exp\left( -\frac{(V_\omega^{-1}(v_t) - \mu_\phi(s_t, a_t))^2}{2\sigma_\phi(s_t, a_t)^2} \right) \right) \Big|_{v_t = \frac{b_t - r(s_t, a_t)}{\gamma}}$.

Example 1 demonstrates that even the simplest MDPs with contrived assumptions on reward, $Q$-function approximators and transition distributions cannot be modelled well by Gaussian models over Bellman operators as the density $p_B(b_t|s_t, a_t, \phi)$ can be arbitrarily far from Gaussian depending upon the choice of function approximator. This issue has been investigated empirically when modelling uncertainty in $Q$-functions (Janz et al., 2019), where improving the representative capacity of a Gaussian model using successor features reduces the learning time from $\mathcal{O}(L^3)$ to $\mathcal{O}(L^{2.5})$ in the $L$-episode length chain task (Osband et al., 2018) under a posterior sampling exploration regime. This issue is particularly pertinent if we are concerned with finding polices that approach Bayes-optimality. Epistemic uncertainty estimates are rendered useless if the space of MDPs that the agent is uncertain over does not reflect the agent's environment. Indeed, as we later prove in Theorem 2 for our proposed method BEN, a model with no capacity for modelling aleatoric uncertainty has a degenerate posterior and the resulting Bayes-optimal policy represents complete exploitation of the current dataset. The key insight is that accurately representing both aleatoric and epistemic uncertainty is crucial for learning Bayesian policies with successful exploration strategies as epistemic uncertainty cannot be considered in isolation from aleatoric uncertainty.

### 4.4 DESIDERATA

In light of the shortcomings of existing BRL approaches presented above, we motivate our approach as satisfying three key desiderata. Our method should:

   I  be a model-free approach to reinforcement learning that allows for bootstrapped samples;

  II  characterise both the epistemic and aleatoric uncertainty in the MDP; and

 III  learn Bayes-optimal policies in the limit of complete optimisation.

## 5 BAYESIAN EXPLORATION NETWORK (BEN)

As we are taking a value based approach in this paper, we focus on solving the optimal Bayesian Bellman equation; however our approach applies equally to the Bayesian Bellman equation for any Bayesian policy. We now derive and introduce the Bayesian Exploration network (BEN), which is comprised of three individual networks: a $Q$-network to reduce the dimensionality of inputs to a one-dimensional variable and then two normalising flow networks to characterise both the aleatoric and epistemic uncertainty over that variable as it passes through the Bellman operator.

### 5.1 RECURRENT $Q$-NETWORK

We introduce a function approximator $Q_\omega : \mathcal{H} \times \mathcal{A} \to \mathbb{R}$ to approximate the optimal Bayesian $Q$-function. Any $\omega^\star \in \Omega$ such that $\mathcal{B}^\star[Q_{\omega^\star}](h_t, a) = Q_{\omega^\star}(h_t, a)$ for observed history $h_t$ and all actions $a \in \mathcal{A}$ thus parametrises an optimal Bayesian $Q$-function from which a Bayes-optimal policy can be derived at each timestep $t$. Similarly to model-free approaches that solve POMDPs (Hausknecht and Stone, 2015; Schlegel et al., 2023), we encode history using a recurrent neural network (RNN). Unlike a myopic approach that solves Eq. (2), our $Q$-function approximator is a mapping from history-action pairs, allowing uncertainty to propagate properly through the Bayesian Bellman equation. In contrast, encoding of history is missing from myopic model-free approaches as uncertainty is only characterised in a single step.

### 5.2 ALEATORIC NETWORK

To characterise the aleatoric uncertainty in the MDP using a model-free approach, we show in Appendix C.1 that the optimal Bayesian Bellman operator acting on $Q_\omega$ can be rewritten as an

expectation over optimal Bellman operators using the posterior $P_\Phi(h_t)$:

$$\mathcal{B}^\star[Q_\omega](h_t, a) = \mathbb{E}_{\phi \sim P_\Phi(h_t)} \left[ \mathbb{E}_{r_t, s_{t+1} \sim P_{R,S}(s_t, a_t, \phi)} \left[ r_t + \gamma \sup_{a'} Q_\omega(h_{t+1}, a') \right] \right]. \qquad (3)$$

Like Fellows et al. (2021), we introduce a random variable $b_t$ using transformation of variables $b_t = r_t + \gamma \sup_{a'} Q_\omega(h_{t+1}, a')$ with distribution $b_t \sim P_B(h_t, a, \phi; \omega)$, which characterises the aleatoric uncertainty in the optimal Bellman operator. More formally, $P_B(h_t, a, \phi; \omega)$ is the *pushforward* distribution satisfying: $\mathbb{E}_{b_t \sim P_B(h_t, a, \phi; \omega)} [f(b_t)] = \mathbb{E}_{r_t, s_{t+1} \sim P_{R,S}(s_t, a_t, \phi)} [f(r_t + \gamma \sup_{a'} Q_\omega(h_{t+1}, a'))]$ for any measurable function $f : \mathbb{R} \to \mathbb{R}$. As samples of $b_t$ are obtained by first sampling $r_t, s_{t+1} \sim P_{R,S}(s_t, a_t, \phi^\star)$ from the underlying MDP $\phi^\star$ then making the transformation $b_t = r_t + \gamma \sup_{a'} Q_\omega(h_{t+1}, a')$, a model of $P_B(h_t, a, \phi; \omega)$ can be learned by using bootstrapped samples thereby satisfying Desideratum I. Given our framework, we now formally investigate the consequence of using a model with no capacity for characterising aleatoric uncertainty:

**Theorem 2.** *Consider a model with degenerate aleatoric uncertainty* $P_B(h_t, a_t, \phi) = \delta(b_t = B(h_t, a_t, \phi))$. *The corresponding posterior is degenerate:* $P_\Phi(h_t) = \delta(\phi = \phi^\star_{MLE}(h_t))$ *and the optimal Bayesian Bellman operator is* $B(h_t, a_t, \phi^\star_{MLE}(h_t))$ *where:* $\phi^\star_{MLE}(h_t) \in \arg\inf_{\phi \in \Phi} \sum_{i=0}^{t-1}(r_i + \gamma \sup_{a' \in \mathcal{A}} Q_\omega(h_{t+1}, a') - B(h_t, a_t, \phi))^2$

As we show in Appendix C, solving the Bellman equation using the optimal Bayesian Bellman operator in $B(h_t, a_t, \phi^\star_{MLE}(h_t))$ recovers an empirical estimate of the temporal difference fixed point (Kolter, 2011; Sutton and Barto, 2018) using a history-conditioned $Q$-function approximator. Theorem 2 demonstrates that even avoiding issues of myopic or contextual approximations outlined in Section 4, a model with no aleatoric uncertainty learns a Bayes-optimal policy with no capacity for exploration.

We use a normalising flow for density estimation (Rezende and Mohamed, 2015; Kobyzev et al., 2021) to model the distribution over optimal Bellman operators $P_B(h_t, a, \phi; \omega)$. Details can be found in Appendix D.1. We refer to this density estimation flow as the *aleatoric* network as it characterises the aleatoric uncertainty in the MDP and its expressiveness implicitly determines the space of MDPs that our model can represent. Unlike in model-based approaches where the hypothesis space must be specified a-priori, in BEN the hypothesis space is determined by the representability of the aleatoric network, which can be tuned to the specific set of problems. Under mild regularity assumptions (Huang et al., 2018), it can be shown that an autoregressive flow as a choice for the aleatoric network can represent any target distribution $P_B(h_t, a, \phi; \omega)$ to arbitrary precision given sufficient capacity and data (Kobyzev et al., 2021), thereby satisfying Desideratum II.

A key advantage of our approach is that we have preprocessed the input to our aleatoric network through the Bayesian $Q$-function approximator $q_t = Q_\omega(h_t, a)$ to extract features that reduce the dimensionality of the state-action space. This architecture hard-codes the prior information that a Bellman operator is a functional of the $Q$-function approximator, meaning that we only need to characterise aleatoric uncertainty in a lower dimensional input $q_t$. Unlike in VariBAD, we do not need to introduce frequentist heuristics to learn function approximator parameters $\omega$. Instead these are learnt automatically by solving the optimal Bayesian Bellman equation, which we detail in Section 5.4.

## 5.3 Epistemic Network

Given a model $P_B(h_t, a_t, \phi; \omega)$, dataset of bootstrapped samples $\mathcal{D}(h_t) := \{b_i\}_{i=0}^{t-1}$ and prior over parameters $P_\Phi$, our goal is to infer the posterior $P_\Phi(\mathcal{D}(h_t))$ to obtain the predictive mean: $\hat{B}[Q_\omega](h_t, a) := \mathbb{E}_{\phi \sim P_\Phi(\mathcal{D}(h_t))} \left[ \mathbb{E}_{b_t \sim P_B(h_t, a_t, \phi; \omega)} [b_t] \right]$ We now prove that the optimal Bayesian Bellman operator is equivalent to the predictive mean, hence BEN is a Bayes-optimal model free approach satisfying Desideratum III.

**Theorem 3.** *Let the transformation of variables* $b_t = r_t + \gamma \sup_{a'} Q_\omega(h_{t+1}, a')$ *be a measurable mapping* $\mathcal{S} \times \mathbb{R} \to \mathbb{R}$ *for all* $\omega \in \Omega, h_t \in \mathcal{H}$. *If* $Q_{\omega^\star}$ *satisfies* $Q_{\omega^\star}(\cdot) = \hat{B}[Q_{\omega^\star}](\cdot)$, *it also satisfies an optimal Bayesian Bellman equation:* $Q_{\omega^\star}(\cdot) = \mathcal{B}^\star[Q_{\omega^\star}](\cdot)$. *Any agent taking action* $a_t \in \arg\sup_a Q_{\omega^\star}(h_t, a)$ *is thus Bayes-optimal with respect to the prior* $P_\Phi$ *and likelihood defined by the model* $P_B(h_t, a_t, \phi; \omega^\star)$.

Unfortunately, inferring the posterior and carrying out marginalisation exactly is intractable for all but the simplest aleatoric networks, which would not have sufficient capacity to represent a complex target

distribution $P_B(h_t, a, \phi; \omega)$. We instead look to variational inference using a parametric normalising flow to learn a tractable approximation $P_\psi$ parametrised by $\psi \in \Psi$ which we learn by minimising the KL-divergence between the two distributions $\text{KL}(P_\psi \parallel P_\Phi(\mathcal{D}(h_t)))$. This is equivalent to minimising the evidence lower bound $\mathcal{L}(\psi; h, \omega)$. We provide details in Appendix D.2. We refer to this flow as the epistemic network as it characterises the epistemic uncertainty in $\phi$. As far as the authors are aware, BEN is the first time flows have been used for combined density estimation and variational inference.

## 5.4 Mean Squared Bayesian Bellman Error (MSBBE)

Having characterised the aleatoric and epistemic uncertainty through BEN, we must learn a parametrisation $\omega^\star$ that satisfies the optimal Bayesian Bellman equation for our $Q$-function approximator. For BEN, this is equivalent to minimising the following Mean Squared Bayesian Bellman Error (MSBBE) between the predictive mean $\hat{B}[Q_\omega](h_t, a)$ and $Q_\omega$: $\text{MSBBE}(\omega; h_t, \psi) := \left\| \hat{B}[Q_\omega](h_t, a) - Q_\omega(h_t, a) \right\|_\rho^2$ where $\rho$ is an arbitrary sampling distribution with support over $\mathcal{A}$. Given sufficient compute, at each timestep $t$ it is possible in principle to solve the nested optimisation problem for BEN: $\omega^\star \in \arg\min_{\omega \in \Omega} \text{MSBBE}(\omega; h_t, \psi^\star(\omega))$ s.t. $\psi^\star(h_t, \omega) \in \arg\min_{\psi \in \Psi} \mathcal{L}(\psi; h, \omega)$. Nested optimisation problems are commonplace in model-free RL and can be solved using two-timescale stochastic approximation: we update the epistemic network parameters $\psi$ using gradient descent on an asymptotically faster timescale than the function approximator parameters $\omega$ to ensure convergence to a fixed point (Borkar, 2008; Heusel et al., 2017; Fellows et al., 2021), with $\omega$ playing a similar role as target network parameters used to stabilise TD.

---

**Algorithm 1** APPROXBRL($P_\Phi, \mathcal{M}(\phi)$)

---
Initialise $\omega, \psi$
Sample initial state $s \sim P_0$
$h = s$
Take $N_{\text{Pretrain}}$ SGD Steps on MSBBE($\omega$)
**while** posterior **not** converged **do**
    Take action $a \in \arg\max_{a'} Q_\omega(h, a')$
    Observe reward $r \sim P_R(s, a, \phi^\star)$
    Transition to new state $s \sim P_S(s, a, \phi^\star)$
    $h \leftarrow \{h, a, r, s\}$
    **for** $N_{\text{Update}}$ Steps: **do**
        Take $N_{\text{Posterior}}$ SGD steps on $\mathcal{L}(\psi; h, \omega)$
        Take a SGD step on MSBBE($\omega; h, \psi$)
    **end for**
**end while**

---

In practice, solving the nested optimisation problem exactly for every observable history $h_t$ is computationally intractable. To avoid issues of computational tractability, we propose partial minimisation of our objectives as outlined in Algorithm 1: after observing a new tuple $\{a, r, s\}$, the agent takes $N_{\text{Update}}$ MSBBE update steps using the new data. This is equivalent to partially minimising the empirical expectation $\mathbb{E}_{h \sim h_t}[\text{MSBBE}(\omega; h, \psi^\star(\omega))]$, where each $h \sim h_t$ is a sequence drawn from the observed history analogously to how state-action pairs are drawn from the replay buffer in DQN (Mnih et al., 2016). To ensure a separation of timescales between parameter updates, the agent carries out $N_{\text{Posterior}}$ steps of stochastic gradient descent on the ELBO for every MSBBE update.

Finally, we exploit the fact that the MSBBE can be minimised prior to learning using samples of state-action pairs and so carry out $N_{\text{Pretrain}}$ pretraining steps of stochastic gradient descent on the loss using the prior in place of the approximate posterior. If no prior knowledge exists, then the agent can be deployed. If there exists additional domain-specific knowledge, such as transitions shared across all MDPs or demonstrations at key goal states, this can also be used to train the agent using the model-based form of the Bellman operator in Eq. (3). Full algorithmic details can be found in Appendix D.3. We remark that BEN's incorporation of prior knowledge doesn't require a full generative model of the environment dynamics and demonstrations can be from simulated or related MDPs that don't exactly match the set of tasks the agent is in at test time.

## 6 Experiments

We introduce a novel search and rescue griworld MDP designed to present a in a high-dimensional challenging extension to the toy tiger problem domain (which we show BEN can solve in Appendix E). An agent is tasked with rescuing $N_{\text{victims}}$ victims from a dangerous situation whilst avoiding any one of $N_{\text{hazards}}$ hazards. Details can be found in Appendix E.4. We evaluate BEN using a $7 \times 7$ gridsize with 8 hazards and 4 victims to rescue.

**Episodic Setting** In the episodic setting, the environment is reset after 245 timesteps and a new environment is uniformly sampled from the space of MDPs. After resetting, the epistemic parameters

$\psi$ are also reset, representing the belief in the new MDP returning to the prior, however the agent maintains it's $Q$-network parameters $\omega$ so it can exploit information that is shared between all MDPs.

We initialise the agent with a zero-mean Gaussian prior of diagonal variance equal to $0.1$ and give the agent no prior domain knowledge except examples of deterministic movement and the average reward for opening a door at random. The results for our implementation are shown in Fig. 1. We plot the return at the end of each 245 timestep episode. As expected, we see that BEN can solve this challenging problem, exploring in initial episodes to learn about how the listening states correlate to victim and hazard positions, then exploiting this knowledge in later episodes, finding all victims immediately. Our results demonstrate that BEN can scale to domains that would be

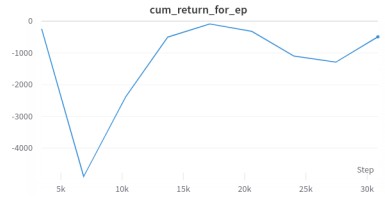

Figure 1: Results of evaluation in search and rescue episodic problem showing the average return of BEN after each episode.

intractable for previous approaches that learn Bayes-optimal policies without forfeiting BRL's strong theoretical properties through approximation.

**Zero-shot Setting** In this setting, our goal is to investigate how effecively BEN can exploit prior knowledge to solve the search and rescue environment in a single episode. We prior-train BEN using simulations in related (but not identical) environments drawn from a uniform prior, showing the agent the affect of listening. Details can be found in Appendix E.5. We plot the cumulative return as a function of number of gradient steps over the course of the episode in Fig. 2 for both BEN and a contextual variant where the $Q$-function approximator has no capacity for representing history.

Our result demonstrates that by exploiting prior knowledge, BEN can successfully rescue all victims and avoid all hazards, even when encountering a novel environment that the agent has never seen a priori. In contrast, existing state-of-the-art model-free methods, which learn a contextual Bayes policy,

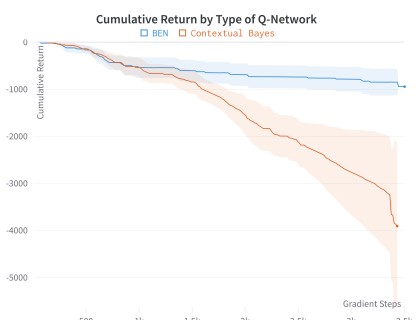

Figure 2: Results of evaluation in zero-shot search and rescue showing BEN vs contextual methods

cannot solve this problem because as our analysis in Section 4 revealed, $\Pi^{\star}_{\text{Contextual}}$ is limited to mixtures of optimal policies conditioned on $\phi$, causing contextual agents to repeatedly hit hazards. This challenging setting showcases the power of our approach, demonstrating high sample efficiency with low computational complexity. Moreover, this setting mimics a real-life application of a search and rescue robot where simulations can be provided by demonstrators in a generic training setting before deployment in a novel environment where the robot has only one chance to succeed.

In addition we performed two ablations; firstly, we demonstrate that performance depends on the capacity of the aleatoric network, verifying our claim in Section 5.2 that there exists a trade-off between specifying a rich enough hypothesis space and a hypothesis space that is too general. Secondly, we investigate how pre-training affects returns. As we decrease the number of prior pre-training MSBBE minimisation steps, we see that performance degrades in the zero-shot settling as expected. Moreover, this ablation shows that a relatively few number of pre-training steps are needed to achieve impressive performance once the agent is deployed in an unknown MDP, supporting our central claim that BEN is computationally efficient Results can be found in Appendix E.7.

## 7 CONCLUSIONS

In this paper we carried out theoretical analyses of existing model-free approaches for BRL, proving that they are limited to optimising over a set of contextual policies or that they optimise a myopic approximation of the true BRL objective. In both cases, the corresponding optimal policies can be arbitrarily Bayes-suboptimal. To correct these issues, we proposed BEN, a model-free BRL approach that can successfully learn true Bayes-optimal policies. Our experimental evaluation confirms our analysis, demonstrating that BEN can behave near Bayes-optimally even under partial minimisation, paving the way for a new generation of model-free BRL approaches with the desirable theoretical properties of model-based approaches.

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

## A  MATHEMATICAL NOTATION

With the exception of policies, we denote probability distributions using uppercase notation $P$ and densities with lower case notation $p$ if they exist. In convention with standard RL notation, we will denote both densities and distributions of policies using $\pi$. The distinction will be clear from context. We use the notation $\delta(x = \mathcal{X}')$ to denote the Dirac distribution centred on the discrete set $\mathcal{X}' \subset \mathcal{X}$, which is defined as:

$$\delta(x = \mathcal{X}') \coloneqq \begin{cases} 1, & x \in \mathcal{X}', \\ 0, & \text{otherwise.} \end{cases}$$

We denote the set of probability measures over a set $\mathcal{X}$ as $\mathcal{P}(\mathcal{X})$. We use Lebesgue integration in the appendix of this paper:

$$\mathbb{E}_{P_X}[f(x)] \coloneqq \int f(x) dP_X(x).$$

We will make use of the $\ell_p$ norms for vector spaces, which for $x \in \mathbb{R}^d$ are defined as:

$$\|x\|_p \coloneqq \left( \sum_{i=1}^{d} |x_i|^p \right)^{\frac{1}{p}}.$$

## B  MODEL-BASED APPROACHES AND THEIR SHORTCOMINGS

Whilst POMDP solvers such as RL$^2$ (Duan et al., 2017) can be naively applied to the BAMDP, solving the BAMDP exactly is hopelessly intractable for all but the simplest of problems for several reasons: firstly, unless a conjugate model is used, the posterior over model parameters cannot be evaluated analytically as the posterior normalisation constant is intractable to evaluate and conjugate models are often too simple to be of practical value in RL; secondly, even if it is possible to obtain a posterior over the model parameters, solving the BAMDP requires evaluating high dimensional integrals by marginalising over model parameters; finally, finding a Bayes-optimal policy in the MDP requires solving a planning problem at each timestep for all possible histories. Few model-based BRL algorithms scale beyond small and discrete state-action spaces, even under approximation (Asmuth and Littman, 2011; Guez et al., 2013).

One notable exception is the VariBAD framework (Zintgraf et al., 2020), and subsequent related approaches (Yao et al., 2021; Zintgraf et al., 2021), which avoids the problem of intractability by being Bayesian over a small subset of model parameters. These methods sacrifice Bayes-optimally, relying on a frequentist heuristic to learn the non-Bayesian parameters; here, parametric models of the reward and state and transition distributions are introduced: $P_\theta(r|s, a, m)$ and $P_\theta(r|s, a, m)$ which are parametrised by $\theta \in \Theta$ and condition on the random variable $m$. A posterior over $m$ is then inferred and used to obtain a marginal likelihood over trajectories:

$$p(\tau|\theta, \mathcal{D}) = \int p_\theta(\tau|m) dP(m|\mathcal{D}),$$

which is optimised for $\theta$. In this way, VariBAD is a partially Bayesian approach, inferring a posterior over $m$, but not parameters $\theta$. The dimensionality of $m$ can be kept relatively small to ensure tractability. As the VariBAD optimal policy is not obtained by marginalising over a space of MDPs nor is the uncertainty accounted for in all of the model's parameters, it is not Bayes-optimal in the limit of approximation.

Another issue with model-based approaches, especially those trained in a meta-learning context such as RL$^2$, is the assumption that the agent has access to a generative hypothesis space where it is possible to sample MDPs from a prior and collect rollouts of transitions from each sampled MDP. In real-world settings, knowing the exact hypothesis space is not a feasible assumption as it is generally not possible to specify transition dynamics a priori for all environments the agent can encounter.

## C  PROOFS

**Lemma 1.** *Let* $\pi \in \Pi_{contextual}$ *with* $\pi(h_t) = \int \pi(s_t, \phi) dP_\Phi(\phi|h_t)$. *It follows:*

$$Q_{\text{Contextual}}^\pi(h_t, a) = \mathbb{E}_{\phi \sim P(\phi|h_t)} \left[ Q^\pi(s_t, a, \phi) \right]. \tag{4}$$

*Proof.*

$$Q^\pi(h_t, a_t) := \mathbb{E}_{\tau_{t:\infty} \sim P_{t:\infty}^\pi(h_t, a_t)} \left[ \sum_{i=t}^\infty \gamma^i r_i \right],$$

$$= \int \left( \sum_{i=t}^\infty \gamma^i r_i \right) dP_{t:\infty}^\pi(\tau_{t:\infty}|h_t, a_t),$$

$$= \int \left( \sum_{i=t}^\infty \gamma^i r_i \right) \int dP_{t:\infty}^\pi(\tau_{t:\infty}|\phi) dP_\Phi(\phi|h_t),$$

$$= \int \int \left( \sum_{i=t}^\infty \gamma^i r_i \right) dP_{t:\infty}^\pi(\tau_{t:\infty}|\phi) dP_\Phi(\phi|h_t),$$

$$= \int Q^\pi(s_t, a_t, \phi) dP_\Phi(\phi|h_t),$$

as required. $\qquad\qquad\qquad\qquad\qquad\qquad\qquad\qquad\qquad\qquad\qquad\qquad\qquad\qquad\qquad\square$

**Theorem 1.** *Contextual Bayesian value functions and optimal policies can be related to frequentist value functions and optimal policies through marginalisation, that is:*

$$\Pi_{\text{Contextual}}^\star = \left\{ \mathbb{E}_{\phi \sim P_\Phi(h_t)} \left[ \pi^\star(\cdot, \phi) \right] | \pi^\star(\cdot, \phi) \in \Pi_\Phi^\star \right\}, \; Q_{\text{Contextual}}^\star(h_t, a) = \mathbb{E}_{\phi \sim P(\phi|h_t)} \left[ Q^\star(s_t, a, \phi) \right].$$

*Proof.* For optimal policies, we first show by contradiction:

$$\Pi_{\text{Contextual}}^\star \supseteq \left\{ \mathbb{E}_{\phi \sim P_\Phi(h_t)} \left[ \pi^\star(\cdot, \phi) \right] | \pi^\star(\cdot, \phi) \in \Pi_\Phi^\star \right\}.$$

Assume that there exists some $\pi^\dagger(\cdot, \phi) \in \Pi_\Phi^\star$ such that $\pi^\dagger(h_t) = \int \pi^\dagger(s_t, \phi) dP_\Phi(\phi|h_t) \notin \Pi_{\text{Contextual}}^\star$. Then by definition, there exists some trajectory-action pair $h_t, a_t$ such that:

$$Q^{\pi^\dagger}(h_t, a_t) < Q^\star(h_t, a_t),$$

$$\implies \int Q^{\pi^\dagger}(s_t, a_t, \phi) dP_\Phi(\phi|h_t) < \int Q^\star(s_t, a_t, \phi) dP_\Phi(\phi|h_t),$$

where the second line follows from Eq. (4) in Lemma 1. This implies that there exists some set $\Phi' \subseteq \Phi$ with non-zero measure according to $P_\Phi(h_t)$ such that:

$$Q^{\pi^\dagger}(s, a, \phi') < Q^\star(s, a, \phi') \, \forall \, \phi' \in \Phi',$$

for some $s, a$ and hence by definition $\pi^\dagger(\cdot, \phi) \notin \Pi_\Phi^\star$, which is a contradiction.

We are left to prove:

$$\Pi_{\text{Contextual}}^\star \subseteq \left\{ \mathbb{E}_{\phi \sim P_\Phi(h_t)} \left[ \pi^\star(\cdot, \phi) \right] | \pi^\star(\cdot, \phi) \in \Pi_\Phi^\star \right\}.$$

Assume that there exists some $\pi^\dagger(\cdot, \phi) \notin \Pi_\Phi^\star$ such that $\pi^\dagger(h_t) = \int \pi^\dagger(s_t, \phi) dP_\Phi(\phi|h_t) \in \Pi_{\text{Contextual}}^\star$. By definition, there exists some $\Phi' \subseteq \Phi$ with non-zero measure according to $P_\Phi(h_t)$ for some $s_t, a_t$ such that:

$$Q^{\pi^\dagger}(s_t, a_t, \phi') < Q^\star(s_t, a_t, \phi') \, \forall \, \phi' \in \Phi',$$

$$\implies \int Q^{\pi^\dagger}(s_t, a_t, \phi) dP_\Phi(\phi|h_t) < \int Q^\star(s_t, a_t, \phi) dP_\Phi(\phi|h_t),$$

$$\implies Q^{\pi^\dagger}(h_t, a_t) < Q^\star(h_t, a_t),$$

which contradicts the definition of $\pi^\dagger(h_t)$, hence

$$\Pi_{\text{Contextual}}^\star = \left\{ \mathbb{E}_{\phi \sim P_\Phi(h_t)} \left[ \pi^\star(\cdot, \phi) \right] | \pi^\star(\cdot, \phi) \in \Pi_\Phi^\star \right\}, , \tag{5}$$

as required.

For the optimal $Q$-function, we start with the definition of the optimal Bayesian Bellman operator:

$$Q^\star(h_t, a_t) = \mathcal{B}^\star[Q^\star](h_t, a_t),$$
$$= \int \left( r_t + \gamma \sup_{a' \in \mathcal{A}} Q^\star(h_{t+1}, a') \right) dP(r_t, s_{t+1} | h_t, a_t),$$
$$= \int r_t dP(r_t | h_t, a_t) + \gamma \int \sup_{a' \in \mathcal{A}} Q^\star(h_{t+1}, a') dP(r_t, s_{t+1} | h_t, a_t).$$

Consider the second term, where we re-write the supremum over actions in terms of the optimal policy for the Bayesian $Q$-function:

$$\int \sup_{a' \in \mathcal{A}} Q^\star(h_{t+1}, a') dP(r_t, s_{t+1} | h_t, a_t)$$
$$= \int \int Q^{\pi^\star}(h_{t+1}, a_{t+1}) d\pi^\star(a_{t+1} | h_{t+1}) dP(r_t, s_{t+1} | h_t, a_t). \tag{6}$$

Using Eq. (4) and Eq. (5) yields:

$$\int \sup_{a' \in \mathcal{A}} Q^\star(h_{t+1}, a') dP(r_t, s_{t+1} | h_t, a_t)$$
$$= \int \int \int Q^{\pi^\star}(s_{t+1}, a_{t+1}, \phi) dP(\phi | h_{t+1}) d\pi^\star(a_{t+1} | h_{t+1}) dP(r_t, s_{t+1} | h_t, a_t),$$
$$= \int \int \int Q^\star(s_{t+1}, a_{t+1}, \phi) d\pi^\star(a_{t+1} | s_{t+1}, \phi) dP(r_t, s_{t+1} | s_t, a_t, \phi) dP(\phi | h_t)$$
$$= \int \int \sup_{a' \in \mathcal{A}} Q^\star(s_{t+1}, a', \phi) dP(r_t, s_{t+1} | s_t, a_t, \phi) dP(\phi | h_t),$$

which we substitute back into Eq. (6):

$$\mathcal{B}^\star[Q^\star](h_t, a_t) = \int r_t dP(r_t | h_t, a_t) + \gamma \int \int \sup_{a' \in \mathcal{A}} Q^\star(s_{t+1}, a', \phi) dP(s_{t+1} | s_t, a_t, \phi) dP_\Phi(\phi | h_t),$$
$$= \int \int \left( r_t + \gamma \sup_{a' \in \mathcal{A}} Q^\star(s_{t+1}, a', \phi) \right) dP(r_t, s_{t+1} | s_t, a_t, \phi) dP_\Phi(\phi | h_t),$$
$$= \int \mathcal{B}^\star[Q^\star](h_t, a_t, \phi) dP_\Phi(\phi | h_t).$$

Finally, substituting for $\mathcal{B}^\star[Q^\star](s_t, a_t, \phi) = Q^\star(s_t, a_t, \phi)$ yields our desired result:

$$Q^\star(h_t, a_t) = \int Q^\star(s_t, a_t, \phi) dP_\Phi(\phi | h_t).$$

$\square$

**Corollary 1.1.** *There exist MDPs with priors such that $\Pi^\star_{Contextual} \cap \Pi^\star_{Bayes} = \varnothing$.*

*Proof.* We consider the tiger problem as a counter example (Kaelbling et al., 1998) with $\gamma = 0.9$, $r_{\text{tiger}} = -500$, $r_{\text{gold}} = 10$ and $r_{\text{listen}} = -1$. Details of the space of MDPs can be found in Appendix E.1. We index the MDP with the tiger in the left door as $\phi = $ tiger left and the tiger in the right door as $\phi = $ tiger right. Consider the uniform prior over MDPs $P(\phi = $ tiger left$) = P(\phi = $ tiger right$) = 0.5$. As agents always start in state $s_0$, it suffices to find the optimal MDP conditioned policies in $s_0$:

$$\pi^\star(s_0, \phi = \text{tiger left}) = \delta(a = \text{open right}), \quad \pi^\star(s_0, \phi = \text{tiger right}) = \delta(a = \text{open left})$$

From Theorem 1, it follows that the optimal Bayesian contextual policy is a mixture of these two policies using the prior:

$$\pi^\star(s_0) = 0.5(\delta(a = \text{open right}) + \delta(a = \text{open left})).$$

The optimal $Q$-function for the optimal MDP-conditioned policies are

$$Q^\star(s_0, a = \text{open right}, \phi = \text{tiger left}) = Q^\star(s_0, a = \text{open left}, \phi = \text{tiger right}) = \frac{r_{\text{gold}}}{1 - \gamma} = 100,$$

$$Q^\star(s_0, a = \text{open right}, \phi = \text{tiger right}) = Q^\star(s_0, a = \text{open left}, \phi = \text{tiger left})$$
$$= r_{\text{tiger}} + \frac{\gamma r_{\text{gold}}}{1 - \gamma} = -155.$$

Using Theorem 1, we can find the Bayesian value function for the optimal contextual policy:

$$Q^\star_{\text{contextual}}(s_0, a = \text{open right})$$
$$= 0.5 \left( Q^\star(s_0, a = \text{open right}, \phi = \text{tiger left}) + Q^\star(s_0, a = \text{open right}, \phi = \text{tiger right}) \right),$$
$$= -27.5,$$
$$Q^\star_{\text{contextual}}(s_0, a = \text{open left})$$
$$= 0.5 \left( Q^\star(s_0, a = \text{open left}, \phi = \text{tiger left}) + Q^\star(s_0, a = \text{open left}, \phi = \text{tiger right}) \right),$$
$$= -27.5,$$

from which the Bayesian return for the optimal contextual policy follows:

$$J^{\pi^\star_{\text{Contextual}}}_{\text{Bayes}} = \mathbb{E}_{s \sim \delta(s_0)} \left[ \mathbb{E}_{a \sim \pi^\star_{\text{Contextual}}(s)} \left[ Q^\star_{\text{contextual}}(s, a) \right] \right],$$
$$= 0.5 \left( Q^\star_{\text{contextual}}(s_0, a = \text{open left}) + Q^\star_{\text{contextual}}(s_0, a = \text{open right}) \right),$$
$$= -27.5.$$

Now consider the policy that always listens $\pi^\dagger = \delta(a = \text{listen})$. The Bayesian return for this policy is:

$$J^{\pi^\dagger}_{\text{Bayes}} = \frac{r_{\text{listen}}}{1 - \gamma} = -10,$$

hence:

$$J^{\pi^\star_{\text{Contextual}}}_{\text{Bayes}} < J^{\pi^\dagger}_{\text{Bayes}} \leq \sup_{\pi \in \Pi_{\mathcal{H}}} J^{\pi}_{\text{Bayes}} = J^{\pi^\star}_{\text{Bayes}},$$
$$\implies \Pi^\star_{\text{Contextual}} \cap \Pi^\star_{\text{Bayes}} = \varnothing,$$

as required. $\qquad \square$

**Example 1.** *Consider the space of MDPs with $\mathcal{S} = \mathbb{R}$, $P_{\mathcal{S}}(s_t, a_t, \phi) = \mathcal{N}(\mu_\phi(s_t, a_t), \sigma_\phi(s_t, a_t))$ and a deterministic reward $r_t = r(s_t, a_t)$ which is known a priori. For any Q-function approximator $Q_\omega(s, a)$ such that $v_t = V_\omega(s_t) := \sup_{a'} Q_\omega(s_t, a')$ with inverse $s_t = V_\omega^{-1}(v_t)$, the distribution over optimal Bellman operators under the transformation $b_t = r(s_t, a_t) + \gamma \sup_{a'} Q_\omega(s_t, a')$ has density:*

$$p_B(b_t | s_t, a_t, \phi) = \left. \left( \frac{\left| \partial_{v_t} V^{-1}(v_t) \right|}{\sqrt{2\pi\sigma_\phi(s_t, a_t)^2}} \exp\left( -\frac{(V_\omega^{-1}(v_t) - \mu_\phi(s_t, a_t))^2}{2\sigma_\phi(s_t, a_t)^2} \right) \right) \right|_{v_t = \frac{b_t - r(s_t, a_t)}{\gamma}}.$$

*Proof.* The result follows immediately by applying the change of variables formula for a random variable. $\qquad \square$

**Theorem 2.** *Consider a model with degenerate aleatoric uncertainty $P_B(h_t, a_t, \phi) = \delta(b_t = B(h_t, a_t, \phi))$. The corresponding posterior is degenerate: $P_\Phi(h_t) = \delta(\phi = \phi^\star_{MLE}(h_t))$ and the optimal Bayesian Bellman operator is $B(h_t, a_t, \phi^\star_{MLE}(h_t))$ where:*

$$\phi^\star_{MLE}(h_t) \in \arg\inf_{\phi \in \Phi} \sum_{i=0}^{t-1} (r_i + \gamma \sup_{a' \in \mathcal{A}} Q_\omega(h_{t+1}, a') - B(h_t, a_t, \phi))^2$$

*Proof.* We characterise the degenerate distribution as the limit of a Gaussian with mean centred on $B(h_t, a_t, \phi)$:

$$P_B(h_t, a_t, \phi) = \delta(b_t = B(h_t, a_t, \phi)) = \lim_{\sigma^2 \to 0} \mathcal{N}(B(h_t, a_t, \phi), \sigma^2).$$

Substituting into the posterior yields:

$$p_\Phi(\phi|h_t) \propto \lim_{\sigma^2 \to 0} \prod_{i=0}^{t-1} \exp\left(-\frac{(b_i - B(h_i, a_i, \phi))^2}{2\sigma^2}\right) p_\Phi(\phi),$$

$$= \lim_{\sigma^2 \to 0} \exp\left(-\sum_{i=0}^{t-1} \frac{(b_i - B(h_i, a_i, \phi))^2}{2\sigma^2} + \log p_\Phi(\phi)\right),$$

$$\implies P_\Phi(h_t) = \delta(\phi = \phi_{\mathrm{MLE}}^\star(h_t)), \quad \phi_{\mathrm{MLE}}^\star(h_t) \in \underset{\phi \in \Phi}{\arg\inf}\left(\sum_{i=0}^{t-1}(b_i - B(h_i, a_i, \phi))^2\right). \quad (7)$$

Recall that the pushforward distribution formally satisfies:

$$\mathbb{E}_{b_t \sim P_B(h_t, a, \phi; \omega)}[f(b_t)] = \mathbb{E}_{r_t, s_{t+1} \sim P_{R,S}(s_t, a_t, \phi)}\left[f\left(r_t + \gamma \sup_{a'} Q_\omega(h_{t+1}, a')\right)\right]. \quad (8)$$

Using the posterior from Eq. (7) and Eq. (8), we derive the corresponding optimal Bayesian Bellman operator as:

$$\mathcal{B}^\star[Q_\omega](h_t, a_t) = \int\int b_t dP_B(b_t|h_t, a_t, \phi)dP_\Phi(\phi|h_t),$$

$$= \int B(h_t, a_t, \phi)d\delta(\phi = \phi_{\mathrm{MLE}}^\star(h_t)),$$

$$= B(h_t, a_t, \phi_{MLE}^\star(h_t)),$$

as required. $\qquad \square$

**Remark:** If we solve the MSBBE using the maximum likelihood (MLE) solution in Eq. (7), we obtain the following solution for $\omega^\star$:

$$\omega^\star \in \underset{\omega \in \Omega}{\arg\min}\|Q_\omega(h_t, a) - B(h_t, a, \phi_{MLE}^\star(h_t))\|_\rho^2 \quad (9)$$

which corresponds to an empirical TD fixed point (Kolter, 2011) using the data $h_t$. Moreover, if we assume further that $Q_\omega(h_t, a)$ and $B(h_t, a, \phi_{MLE}^\star(h_t))$ share a function space, that is there exists and $\omega_t^\star$ for every $\phi_{MLE}^\star(h_t)$ such that $B(h_t, \cdot, \phi_{MLE}^\star(h_t)) = Q_{\omega_t^\star}(h_t, \cdot)$, then we can minimise the MSBBE in Eq. (9) by setting $\omega_t^\star = \phi_{MLE}^\star(h_t)$. Taking $k$ gradient descent steps to find $\phi_{MLE}^\star(h_t)$ while keeping $\omega_t^\star$ fixed:

$$\phi \leftarrow \phi - \alpha(r_i + \gamma \sup_{a \in \mathcal{A}} Q_{\omega_t^\star}(h_{i+1}, a) - B(h_i, a_i, \phi))\nabla_\phi B(h_i, a_i, \phi)$$

and periodically updating the target parameters to $\omega_{t+1}^\star = \phi$ then becomes exactly equivalent to using a target network to stabilise TD (Mnih et al., 2015; Fellows et al., 2023).

## C.1 THE SUFFICIENCY PRINCIPLE

In the context of Bayesian RL, the sufficiency principle is characterised as follows: let $f_t = F(s_{t+1}, r_t)$ be a sufficient statistic for $\phi$. The sufficiency principle states that the same inference about $\phi$ can be made by observing any sufficient statistic $f_t$ as directly observing $s_{t+1}, r_t$ (Birnbaum, 1962). In the case of BEN, the agent observe samples of the variable $b_t$, which is a sufficient statistic for $\phi$ if we can learn a parametrisation that characterises and optimal Bellman operator: as each MDP has a unique optimal Q-function and corresponding optimal Bellman operator, the mapping from $\Phi$ to the space of Bellman operators is one-to-one. Similarly, each MDP is uniquely defined by its set of transition distributions, and so the mapping from $\Phi \to \mathcal{S} \times \mathbb{R}$ is also one-to-one. Overall the transformation of variables $b_t = r_t + \gamma \sup_{a'} Q^\star(h_{t+1}, a')$ is thus one-to-one as each unique transition distribution has a unique optimal Bellman operator, hence $b_t$ must be sufficient variable for learning $\phi^\star$.

From this perspective, it does not matter whether we choose to take a model-free approach, characterising uncertainty in the optimal Bellman operator, or a model-based approach, characterising uncertainty in the transition distributions, a Bayes-optimal policy may still be learned.

**Theorem 3.** *Let the transformation of variables $b_t = r_t + \gamma \sup_{a'} Q_\omega(h_{t+1}, a')$ be a measurable mapping $\mathcal{S} \times \mathbb{R} \to \mathbb{R}$ for all $\omega \in \Omega, h_t \in \mathcal{H}$. If $Q_{\omega^\star}$ satisfies $Q_{\omega^\star}(\cdot) = \hat{B}[Q_{\omega^\star}](\cdot)$, it also satisfies an optimal Bayesian Bellman equation: $Q_{\omega^\star}(\cdot) = \mathcal{B}^\star[Q_{\omega^\star}](\cdot)$. Any agent taking action $a_t \in \arg\sup_a Q_{\omega^\star}(h_t, a)$ is thus Bayes-optimal with respect to the prior $P_\Phi$ and likelihood defined by the model $P_B(h_t, a_t, \phi; \omega^\star)$.*

*Proof.* As $Q_{\omega^\star}$ satisfies $Q_{\omega^\star}(\cdot) = \hat{B}[Q_\omega](\cdot)$, and from the definition of the normalising flow defining the aleatoric network, we can write for every $t$:

$$Q_{\omega^\star}(h_t, a_t) = \int \mathbb{E}_{z_{\mathrm{al}} \sim P_{\mathrm{al}}} \left[ B(q_t, z_{\mathrm{al}}, \phi) \right] dP_\Phi(\phi | \mathcal{D}(h_t)),$$

$$= \int \int b_t dP_B(h_t, a_t, \phi; \omega) dP_\Phi(\phi | \mathcal{D}(h_t)).$$

As $b_t$ is defined as a measurable transformation of variables $b_t = r_t + \gamma \sup_{a'} Q_{\omega^\star}(h_{t+1}, a')$, it follows from the change of variables theorem under measurable mappings (see Bogachev (2007, Theorem 3.6.1)):

$$\int b_t dP_B(h_t, a_t, \phi; \omega) = \int \int \left( r_t + \gamma \sup_{a'} Q_{\omega^\star}(h_{t+1}, a') \right) dP_S(s_{t+1} | s_t, a, \phi) dP_R(r_t | s_t, a, \phi),$$

hence:

$$Q_{\omega^\star}(h_t, a_t) = \mathcal{B}^\star[Q_{\omega^\star}](h_t, a_t),$$

$$= \int \left( \int \int \left( r_t + \gamma \sup_{a'} Q_{\omega^\star}(h_{t+1}, a') \right) dP_S(s_{t+1} | s_t, a_t, \phi) dP_R(r_t | s_t, a_t, \phi) \right) dP_\Phi(\phi | h_t),$$

$$= \int \int \left( r_t + \gamma \sup_{a'} Q^\star(h_{t+1}, a') \right) \int dP_S(s_{t+1} | s_t, a_t, \phi) dP_R(r_t | s_t, a_t, \phi) dP_\Phi(\phi | h_t),$$

$$= \int \int \left( r_t + \gamma \sup_{a'} Q^\star(h_{t+1}, a') \right) dP_{R,S}(s_{t+1}, r_t | h_t, a_t),$$

$$= \int \left( r_t + \gamma \sup_{a'} Q^\star(h_{t+1}, a') \right) dP_{\mathcal{H}}(h_{t+1} | h_t, a_t), \tag{10}$$

where we have used the fact that $b_t$ is a sufficient variable for learning $\phi$ to exchange the posterior given observations $\mathcal{D}(h_t)$ for the posterior given $h_t$, as the sufficiency principle ensures that the same inference about $\phi$ is made by observing either $b_t$ or $s_{t+1}, r_t$ (Birnbaum, 1962). Eq. (10) is exactly the optimal Bayesian Bellman operator from the definition in Eq. (1), hence $Q_{\omega^\star}(\cdot)$ satisfied the optimal Bayesian Bellman equation for the BAMDP defined by model $P_S(s_{t+1} | s_t, a_t, \phi)$ and $P_R(r_{t+1} | s_t, a_t, \phi)$ and prior $P_\Phi$. As $Q_{\omega^\star}(\cdot)$ satisfies the optimal Bayesian Bellman equation, it is an optimal Bayesian $Q$-function, so by definition, any $a_t \in \arg\sup_a Q_{\omega^\star}(h_t, a)$ is an action taken by a Bayes-optimal agent. $\square$

## D  NETWORK DETAILS

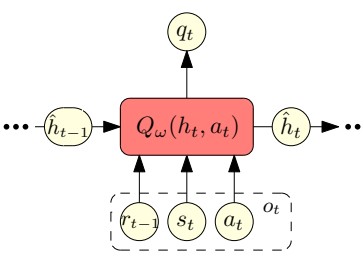

Figure 3: Recurrent $Q$ Network

Our recurrent $Q$ network architecture is shown Fig. 3. We encode the prior history-action pair via a recurrent variable $\hat{h}_t := \{h_t, a_t\}$. At each timestep our network inputs $\hat{h}_{t-1}$ and a tuple of observations $o_t := \{r_{t-1}, s_t, a_t\}$ and outputs the value variable $q_t = Q_\omega(h_t, a_t) = Q_\omega(\hat{h}_{t-1}, o_t)$ and the new recurrent encoding $\hat{h}_t$.

### D.1  ALEATORIC NETWORK

To model the distribution over optimal Bellman operators $P_B(h_t, a, \phi; \omega)$ we introduce a base variable $z_{\mathrm{al}} \in \mathbb{R}$ with a tractable distribution $P_{\mathrm{al}}$; in this paper we use a zero-mean, unit variance Gaussian $\mathcal{N}(0, 1)$. We then generate $b_t$ using a change of variables $b_t = B(z_{\mathrm{al}}, q_t, \phi)$ parameterised by $\phi \in \Phi$, where $B(z_{\mathrm{al}}, q_t, \phi)$ is a mapping in $z_{\mathrm{al}}$ with inverse $z_{\mathrm{al}} = B^{-1}(b_t, q_t, \phi)$. Under this change of variables, $\mathbb{E}_{b_t \sim P_B(h_t, a, \phi; \omega)} [f(b_t)] =$

$\mathbb{E}_{z_{\text{al}} \sim P_{\text{al}}} [f \circ B(z_{\text{al}}, q_t, \phi)]$ for any integrable $f : \mathbb{R} \to \mathbb{R}$. We refer to $B(z_{\text{al}}, q_t, \phi)$ as the *aleatoric* network as it characterises the aleatoric uncertainty in the MDP and its expressiveness implicitly determines the space of MDPs that our model can represent.

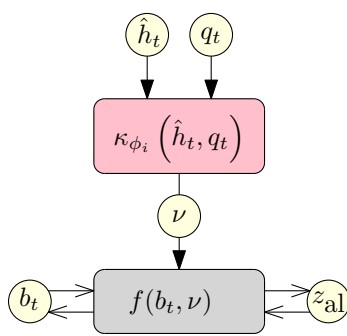

Figure 4: Details of Aleatoric Flow

We define our aleatoric flow by adapting the autoregressive flow (Kingma et al., 2016). We take inputs from the RNN $Q$-function approximator outputs $\hat{h}_t, q_t$ (including the history encoding) and pass them through a conditioner $\kappa_{\phi_i}(\hat{h}_t, q_t)$, which is a feed forward neural network parametrised by $\phi_i$ where $\phi_i \subset \phi$. The output of the conditioner is a vector that defines the parameters for the coupling function. We use an inverse autoregressive flow, with $z_{\text{al}} \in \mathbb{R}^2$ followed with a dimensionality reduction layer to reduce the dimension of the output to 1 and abs layers, as detailed in Nielsen et al. (2020). Since only $b_t = B(q_t, z_{\text{al}}, \phi)$ needs to be bijective in $z_{\text{al}}$, there are also no restrictions on $Q_\omega(h_t, a)$, allowing us to use any arbitrary RNN. The aleatoric network then consists of $L$ coupling functions in composition:

$$B(z_{\text{al}}, q_t, \phi) = f_L \circ f_{L-1} \circ \cdots f_2 \circ f_1(z_{\text{al}}, \hat{h}_t, q_t, \phi).$$

## D.2 EPISTEMIC NETWORK

Given the aleatoric network $B(z_{\text{al}}, q_t, \phi)$, dataset of bootstrapped samples $\mathcal{D}(h_t) \coloneqq \{b_i\}_{i=0}^{t-1}$ and prior over parameters $P_\Phi$, our goal is to infer the posterior $P_\Phi(\mathcal{D}(h_t))$ to obtain the predictive mean:

$$\hat{B}[Q_\omega](h_t, a) \coloneqq \mathbb{E}_{\phi \sim P_\Phi(\mathcal{D}(h_t))} \left[ \mathbb{E}_{z_{\text{al}} \sim P_{\text{al}}} [B(q_t, z_{\text{al}}, \phi)] \right].$$

Unfortunately, inferring the posterior and carrying out marginalisation exactly is intractable for all but the simplest aleatoric networks, which would not have sufficient capacity to represent a complex target distribution $P_B(h_t, a, \phi; \omega)$. We instead look to variational inference using a normalising flow to learn a tractable approximation.

Like in Appendix D.1, we introduce a base variable $z_{\text{ep}} \in \mathbb{R}^d$ with a tractable distribution $P_{\text{ep}}$; again we use a zero-mean Gaussian $\mathcal{N}(0, I^d)$. We then make a transformation of variables $\phi = t_\psi(z_{\text{ep}})$ where $t_\psi : \mathbb{R}^d \to \mathbb{R}^d$ is a bijective mapping parametrised by $\psi \in \Psi$ with inverse $z_{\text{ep}} = t_\psi^{-1}(\phi)$. We refer to $t_\psi(z_{\text{ep}})$ as the epistemic network as it characterises the epistemic uncertainty in $\phi$. From the change of variables formula, it follows that the resulting variational distribution $P_\psi$ has a density $p_\psi(\phi) = |\det (J_\psi(\phi))| \, p_{\text{ep}} \circ t_\psi^{-1}(\phi)$ where $J_\psi(\phi) \coloneqq \nabla_\phi t_\psi^{-1}(\phi)$ is the Jacobian of the inverse mapping. Using variational inference, we treat $P_\psi$ as an approximation of the true posterior $P_\Phi(h_t)$, which we learn by minimising the KL-divergence between the two distributions $\text{KL}(P_\psi \parallel P_\Phi(h_t))$. This is equivalent to minimising the following negative evidence lower-bound (ELBO) objective with respect to $\psi$:

$$\mathcal{L}(\psi; h_t, \omega) \coloneqq \mathbb{E}_{z_{\text{ep}} \sim P_{\text{ep}}} \left[ \left( \sum_{i=0}^{t-1} \left( B^{-1}(b_i, q_i, \phi)^2 - \log \left| \partial_b B^{-1}(b_i, q_i, \phi) \right| \right) - \log p_\Phi(\phi) \right) \Bigg|_{\phi = t_\psi(z_{\text{ep}})} \right].$$

We provide a fully connected schematic of BEN with flow details in Fig. 5.

To derive this result, we start with the definition of the KL-divergence $\text{KL}(P_\psi \parallel P_\Phi(h_t))$, using Bayes' rule to re-write the log-posterior:

$$\text{KL}(P_\psi \parallel P_\Phi(h_t)) \coloneqq \int \left( \log p_\psi(\phi) - \log p_\Phi(\phi | h_t) \right) dP_\psi(\phi),$$

$$= \int \left( \log p_\psi(\phi) - \log p_\Phi(h_t | \phi) - \log p_\Phi(\phi) + \log p(h_t) \right) dP_\psi(\phi),$$

$$= \int \left( \log p_\psi(\phi) - \log p_\Phi(h_t | \phi) - \log p_\Phi(\phi) \right) dP_\psi(\phi) + \log p(h_t).$$

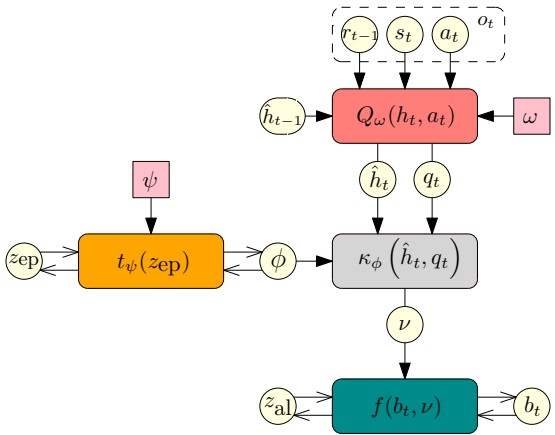

Figure 5: Schematic of BEN

As the log-evidence $\log p(h_t)$ has no dependence on $\psi$, we can omit it from our objective, instead maximising the ELBO:

$$\mathcal{L}(\psi; h_t, \omega) = \int \left( \log p_\Phi(h_t|\phi) + \log p_\Phi(\phi) - \log p_\psi(\phi) \right) dP_\psi(\phi).$$

Using Eq. (8), we write the expectation with respect to $P_\psi$ using the transformation of variables:

$$\mathcal{L}(\psi; h_t, \omega) = \int \left( \log p_\Phi(h_t|\phi) + \log p_\Phi(\phi) - \log p_\psi(\phi) \right)|_{\phi=t_\psi(z_{\mathrm{ep}})} dP_{\mathrm{ep}}(z_{\mathrm{ep}}),$$
$$= \int \left( \log p_\Phi(h_t|\phi) + \log p_\Phi(\phi) \right)|_{\phi=t_\psi(z_{\mathrm{ep}})} dP_{\mathrm{ep}}(z_{\mathrm{ep}}) - \int \log p_\psi \circ t_\psi(z_{\mathrm{ep}}) dP_{\mathrm{ep}}(z_{\mathrm{ep}}).$$

Now, by the definition of $p_\psi(\phi)$, it follows:

$$p_\psi \circ t_\psi(z_{\mathrm{ep}}) = |\det \left( J_\psi \circ t_\psi(z_{\mathrm{ep}}) \right)| \, p_{\mathrm{ep}} \circ t_\psi^{-1} \circ t_\psi(z_{\mathrm{ep}}),$$
$$= \left| \det \left( \nabla_\phi t_\psi^{-1} \circ t_\psi(z_{\mathrm{ep}}) \right) \right| p_{\mathrm{ep}}(z_{\mathrm{ep}}),$$
$$= p_{\mathrm{ep}}(z_{\mathrm{ep}}).$$

As $p_{\mathrm{ep}}(z_{\mathrm{ep}})$ has no dependence on $\psi$, we can omit it from the objective, yielding:

$$\mathcal{L}(\psi; h_t, \omega) = \int \left( \log p_\Phi(h_t|\phi) + \log p_\Phi(\phi) \right)|_{\phi=t_\psi(z_{\mathrm{ep}})} dP_{\mathrm{ep}}(z_{\mathrm{ep}}),$$
$$= \int \left( \log \left( \prod_{i=0}^{t-1} p_B(b_i|h_i, a_i, \phi; \omega) \right) + \log p_\Phi(\phi) \right) \Bigg|_{\phi=t_\psi(z_{\mathrm{ep}})} dP_{\mathrm{ep}}(z_{\mathrm{ep}}),$$
$$= \int \left( \sum_{i=0}^{t-1} \log p_B(b_i|h_i, a_i, \phi; \omega) + \log p_\Phi(\phi) \right) \Bigg|_{\phi=t_\psi(z_{\mathrm{ep}})} dP_{\mathrm{ep}}(z_{\mathrm{ep}}).$$

Finally we can derive the exact form of the log-density $p_B(b_i|h_i, a_i, \phi; \omega)$ using the change of variables formula under $b_t = B(z_{\mathrm{al}}, q_t, \phi)$:

$$\log p_B(b_i|h_i, a_i, \phi; \omega) = \log \left( \exp(-B^{-1}(b_i, q_i, \phi)^2) \left| \partial_b B^{-1}(b_i, q_i, \phi) \right| \right),$$
$$= -B^{-1}(b_i, q_i, \phi)^2 + \log \left| \partial_b B^{-1}(b_i, q_i, \phi) \right|.$$

Substituting and multiplying by $-1$, thus changing to an objective to minimise rather than maximise, yields our desired result:

$$\mathcal{L}(\psi; h_t, \omega) := \mathbb{E}_{z_{\mathrm{ep}} \sim P_{\mathrm{ep}}} \left[ \left( \sum_{i=0}^{t-1} \left( B^{-1}(b_i, q_i, \phi)^2 - \log \left| \partial_b B^{-1}(b_i, q_i, \phi) \right| \right) - \log p_\Phi(\phi) \right) \Bigg|_{\phi=t_\psi(z_{\mathrm{ep}})} \right].$$

### D.3   NETWORK TRAINING

**Algorithm 2** PRIORINITIALISATION($P_\Phi, s_0, \mathcal{D}_{\text{prior}}$)

> Initialise $\omega$
> **for** $N_{\text{Pretrain}}$ steps **do**
>   Sample action $a \sim \rho$
>   Sample two MDPs $\phi, \phi' \sim P_\Phi$
>   Sample aleatoric variables $z_{\text{al}}, z_{\text{al}}' \sim P_{\text{al}}$
>   $q_0 = Q_\omega(s_0, a)$
>   $\omega \leftarrow \omega - \alpha(B(q_0, z_{\text{al}}, \phi) - q_0)\nabla_\omega(B(q_0, z_{\text{al}}', \phi') - q_0)$
>   Sample from prior data $s, a \sim \mathcal{D}_{\text{prior}}$
>   Sample two MDPs $\phi, \phi' \sim P_\Phi$
>   Sample rewards-state transition $r, s_+ \sim P_{R,S}(s, a, \phi)$
>   Sample rewards-state transition $r', s_+' \sim P_{R,S}(s, a, \phi')$
>   $\omega \leftarrow \omega - \alpha(r + \gamma \sup_{a' \in \mathcal{A}} Q_\omega(s, a, r, s_+, a') - Q_\omega(s, a))\nabla_\omega(\gamma \sup_{a' \in \mathcal{A}} Q_\omega(s, a, r', s_+', a') - Q_\omega(s, a))$
> **end for**

**Prior Initialisation** Before any actions have been taken, we can minimise the MSBBE using the initial state $s_0$ and the prior $P_\phi$, which we assume is tractable to obtain samples from. This has the advantage of initialising the $Q$-function approximator to incorporate any prior domain knowledge we have about the MDP, in addition to ensuring that an optimal Bayesian Bellman equation is approximately satisfied before training starts. Once the posterior is updated using a new observation, we shouldn't expect the solution to the MSBBE to change significantly to reflect the updated belief. Finally, there may be prior knowledge about state and reward transitions that are available to us a priori that we would like to encode in the $Q$-function approximator. If an agent in state $s$ taking action $a$ always transitions according to a known conditional distribution $P_{R,S}(s, a, \phi)$, then we can use this information to solve the Bayesian Bellman equation conditioned on $s, a$. We combine all such state-action pairs into a dataset $\mathcal{D}_{\text{prior}} \coloneqq \{s_i, a_i\}_{i=1}^{K_{\text{prior}}}$, for which we minimise the MSSBE:

$$\mathcal{L}(\omega; \mathcal{D}_{\text{prior}})$$
$$= \sum_{i=1}^{K_{\text{prior}}} \left( \mathbb{E}_{\phi \sim P_\Phi} \left[ \mathbb{E}_{r_i, s_{i+1} \sim P_{R,S}(s_i, a_i, \phi)} \left[ r + \gamma \sup_{a' \in \mathcal{A}} Q_\omega(s_i, a_i, r_i, s_{i+1}, a') \right] \right] - Q_\omega(s_i, a_i) \right)^2.$$

We give specific details of $\mathcal{D}_{\text{prior}}$ in the context of our search and rescue environment in Appendix E.4. Both MSBBE objectives can be minimised using stochastic gradient descent with two independent samples from the prior to avoid bias in our updates, as outlined in Algorithm 2. Note that for domains where we don't have such knowledge, we can take $\mathcal{D}_{\text{prior}} = \varnothing$ and ignore the minimisation steps on $\mathcal{L}(\omega; \mathcal{D}_{\text{prior}})$.

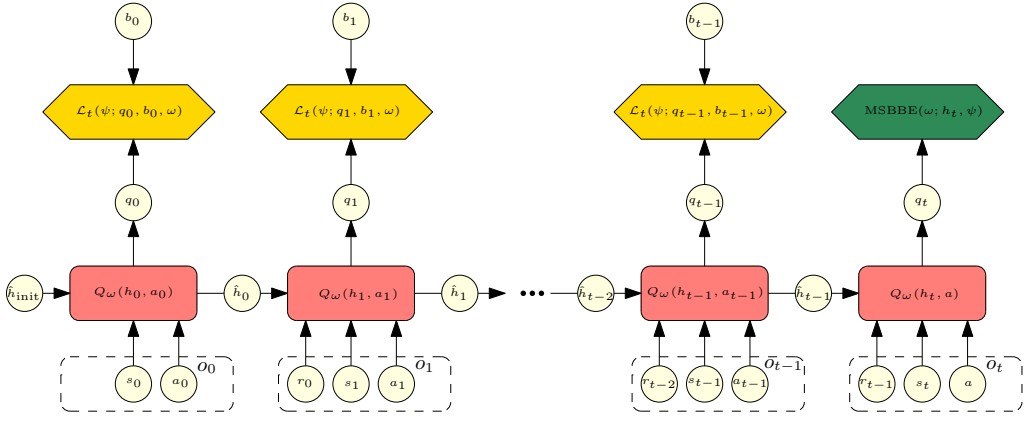

Figure 6: Schematic of BEN Training Regime. Losses are shown as hexagons.

**Posterior Updating** To obtain an efficient algorithm, we note that the ELBO objective can be written as a summation: $\mathcal{L}(\psi; h_t, \omega) = \sum_{i=0}^{t-1} \mathcal{L}_t(\psi; q_i, b_i, \omega)$, where each sub-objective is:

$$\mathcal{L}_t(\psi; q_i, b_i, \omega) \coloneqq \mathbb{E}_{z_{\text{ep}} \sim P_{\text{ep}}} \left[ \left( B^{-1}(b_i, q_i, \phi) \right)^2 - \log \left| \partial_b B^{-1}(b_i, q_i, \phi) \right| - \frac{1}{t} \log p_\Phi(\phi) \Big|_{\phi = t_\psi(z_{\text{ep}})} \right].$$

As shown in Fig. 6, we can minimise $\mathcal{L}(\psi; h_t, \omega)$ by unrolling the RNN, starting at $i = 0$. After each timestep, we obtain $q_i$, which can be used to minimise the loss $\mathcal{L}_t(\psi; q_i, b_i, \omega)$ with the observation $b_i$ whilst keeping $\omega$ fixed. Once the network has been unrolled to the timestep $t$, we can use the output to minimise the MSBBE. Like in for our prior initialisation algorithm in Algorithm 2, it is important that we sample two independent samples $\phi, \phi' \sim P_\phi(h_t)$ from our approximate posterior when minimising the MSBBE to avoid biased gradient estimates. Once $t$ becomes too large, we can truncate the sequences to length $t'$, starting at state $s_{t-t'}$ instead of $s_0$. Like when target parameters used to stabilise frequentist TD methods (Fellows et al., 2023), this updating ensures that the $Q$-network is updated on an asymptotically slower timescale to the posterior parameters, and we tune the length of truncation $t'$ for the sequence and stepsizes $\alpha_\psi$ and $\alpha_\omega$ to ensure stability.

## E EXPERIMENTS

### E.1 TIGER PROBLEM

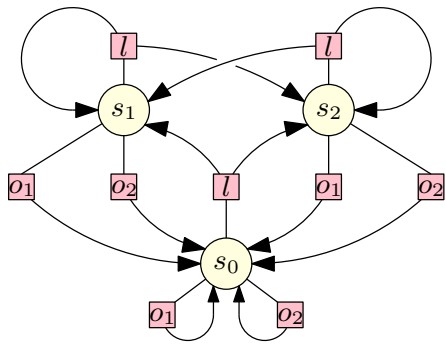

Figure 7: Tiger Problem MDP

**Algorithm 3** POSTERIORUPDATING$(h_t, \psi, \omega)$

> **for** $N_{\text{Update}}$ steps **do**
>     $\hat{h} \leftarrow \hat{h}_{\text{init}}$
>     $o \leftarrow \{s_0, a_0\}$
>     **for** $i \in [0 : t-1]$ **do**
>         $\hat{h}, q_i \leftarrow Q_\omega(\hat{h}, o)$
>         $b_i \leftarrow r_i + \gamma \sup_{a'} Q_\omega(\hat{h}, o, r_i, s_{i+1}, a')$
>         $g_\psi \sim \nabla_\psi \mathcal{L}_t(\psi; q_i, b_i, \omega)$
>         $\psi \leftarrow \psi - \alpha_\psi g_\psi$
>         $o \leftarrow r_i, s_{i+1}, a_{i+1}$
>     **end for**
>     $q_t \leftarrow Q_\omega(\hat{h}, o)$
>     $g_\omega \sim \nabla_\omega \text{MSBBE}(\omega; q_t, \psi)$
>     $\omega \leftarrow \omega - \alpha_\omega g_\omega$
> **end for**

The aim of this empirical evaluation is to verify our claim that BEN can learn a Bayes-optimal policy and compare BEN to existing model-free approaches. We evaluate BEN in the counterexample tiger problem domain from Corollary 1.1, which allows for comparison against a true Bayes-optimal policy. We show our tiger problem MDP in Fig. 7. The agent is always initialised in state $s_0$ and can chose to open door 1 ($o_1$), open door 2 ($o_2$) or to listen ($l$): $\mathcal{A} := \{o_1, o_2, l\}$. There are two possible MDPs the agent can be in, with the tiger assigned to either door 1 or door 2 randomly and the gold to the other door. If the agent chooses $o_1$, door 1 is opened and the agent receives a reward of $r_{\text{tiger}} = -500$ if the tiger is behind the door or $r_{\text{gold}} = 10$ if the gold is behind the door. The agent always transitions to state $s_0$ after selecting $o_1$ or $o_2$. If the agent chooses to listen, it receives a small negative reward of $r_{\text{listen}} = -1$ and if the tiger is behind door 1 transitions to state $s_1$ with probability 0.85 and state $s_2$ with probability 0.1, or if the tiger is behind door 2, the agent transitions to state $s_2$ with probability 0.85 and state $s_1$ with probability 0.1.

### E.2 TIGER PROBLEM IMPLEMENTATION DETAILS

We initialise the agent with a uniform prior over the two MDPs. The posterior for this problem is tractable so we use that in place of the epistemic network:

$$P_\Phi(\phi = \text{tiger in } 1|h_t) = \frac{0.85^{N_1} \cdot 0.1^{N_2}}{0.85^{N_1} \cdot 0.1^{N_2} + 0.1^{N_1} \cdot 0.85^{N_2}},$$

$$P_\Phi(\phi = \text{tiger in } 2|h_t) = 1 - P_\Phi(\phi = \text{tiger in } 1)$$

where $N_1$ is the number of visitations to state $s_1$ and $N_2$ is the number of visitations to state $s_2$. If the agent opens the door, the posterior trivially becomes:

$$P_\Phi(\phi|h_t) = \begin{cases} \delta(\phi = \text{tiger in 1}), & a_{t-1} = o_1, r_{t-1} = r_{\text{tiger}}, \\ \delta(\phi = \text{tiger in 1}), & a_{t-1} = o_2, r_{t-1} = r_{\text{gold}}, \\ \delta(\phi = \text{tiger in 2}), & a_{t-1} = o_2, r_{t-1} = r_{\text{tiger}}, \\ \delta(\phi = \text{tiger in 2}), & a_{t-1} = o_1, r_{t-1} = r_{\text{gold}}. \end{cases}$$

The aleatoric network can be handcoded as the pushforward of known transition distributions. We vary the number of steps for the MSBBE minimisation with a learning rate of 0.02 using ADAM for the stochastic gradient descent. For the for Q-function approximator, we use a fully connected linear layer with ReLU activations, a gated recurrent unit and a final fully connected linear layer with ReLU activations. All hidden dimensions are 32. The dimension of $\hat{h}_0$ is 2. The input dimension is 1 + 2 =3 and the network output is 3 dimensional to reflect the three possible actions the agent can take.

### E.3 TIGER PROBLEM RESULTS

We initialise all agents in a tabula rasa setting with a uniform prior over MDPs and plot the median returns after each timestep in Fig. 8 for 11 timesteps, averaged over MDPs, each drawn uniformly. We plot the performance of BEN for a varying number of SGD minimisation steps on our MSBBE objective. Fig. 8 shows that by increasing the number of SGD minimisation steps, BEN's performance approaches that of the Bayes-optimal oracle and the variance in the policies decreases, with near Bayes-optimal performance attainted using 20 minimisation steps. We also compare BEN to an oracle that is optimal over the space contextual policies, $\Pi^\star_{\text{Contextual}}$, which is an optimal policy for existing model-free approaches. As expected, the contextual optimal policy is limited to a mixture of optimal policies conditioned on $\phi$, hence the performance is comparatively poor: median returns are significantly lower than BEN as contextual policies sample an initial action uniformly before acting optimally once the true MDP is revealed.

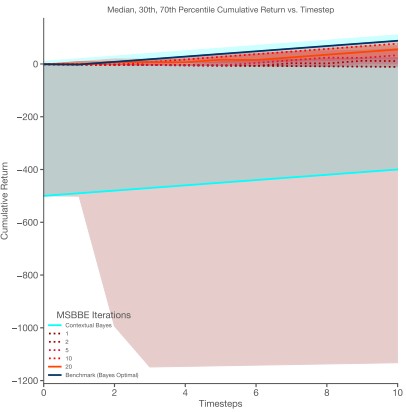

Figure 8: Results of evaluation in tiger problem showing BEN with increasing minimisation steps on MSBBE vs Bayes-optimal and contextual oracles

### E.4 SEARCH AND RESCUE PROBLEM

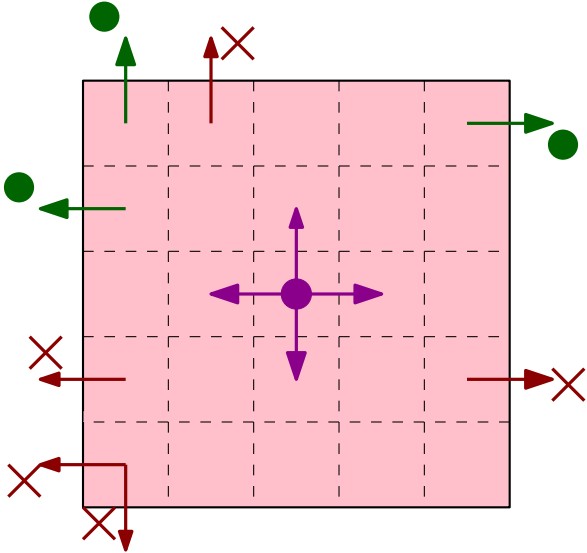

Figure 9: $5 \times 5$ Search and Rescue Problem MDP with 5 Hazards (red crossed) and 3 Victims (green circles). Agent (purple circle) is shown in $s_0$. Green actions yield reward $r_{\text{rescue}}$ and red actions yield reward $r_{\text{hazard}}$.

We now present a novel search and rescue MDP designed to present a challenging extension to the toy tiger problem domain. An agent is tasked with rescuing $N_{\text{victims}}$ victims from a dangerous situation whilst avoiding any one of $N_{\text{hazards}}$ hazards. The agent's action space is $\mathcal{A} = \{up, down, left, right, listen\}$. The agent can move in an $N_{\text{grid}} \times N_{\text{grid}}$ gridworld where $N_{\text{grid}}$ is an odd number and transitions one square deterministically in the direction of the action taken. If the agent selects an action that would take it off the grid, it remains put and opens the door adjacent to its square in the direction of the action. If the agent opens a door with a victim behind, it receives a reward of $r_{\text{victim}} = 10$ and the victim is removed from the MDP. If the agent . If the agent opens a door with a hazard behind, it receives a reward of $r_{\text{hazard}} = -100$ and the hazard remains in the MDP. We show an example MDP in Fig. 9.

The agent is initialised in position $(0,0)$, which is the central square of the grid. The agent observes state $s \in \mathcal{S} \subset \mathbb{R}^{2+N_{\text{victims}}+N_{\text{hazards}}}$ where $l_{\text{agent}} := (s_0, s_1)$ is the agent's location relative to $(0,0)$. The agent does not directly observe which doors have hazards or victims behind. If the agent chooses the action listen, their location remains put and they transition to a new state $s'$ where $s'_i$ for each $i \in \{2 : 1 + N_{\text{victims}} + N_{\text{hazards}}\}$ is given by:

$$s'_i = \exp\left(-\frac{\|l_{\text{agent}} - l_i\|^2}{N_{\text{grid}}} + \eta\right), \quad \eta \sim \mathcal{N}(0; \sigma^2_{\text{noise}})$$

which a noisy variable correlated to the distance between the agent and each victim/hazard $l_i$. The victim locations are $\{l_i\}_{i \in \{2:N_{\text{victims}}+1\}}$ and the hazard locations are $\{l_i\}_{i \in \{N_{\text{victims}}+2:1+N_{\text{victims}}+N_{\text{hazard}}\}}$. For each MDP, the victims and hazards are randomly assigned a square each adjacent to the grid and the initialised uniformly across that square. If an agent opens a door with a victim, their location becomes $(N_{\text{grid}} \cdot 1000, N_{\text{grid}} \cdot 1000)$ and no further reward can be obtained for that victim. Agents receive a small negative reward for listening $r_{\text{listen}} = -1$ and no reward for traversing the grid.

### E.5 Exploiting Prior Knowledge

For the search and rescue environment, there is domain knowledge that we can use to form $\mathcal{D}_{\text{prior}}$ that is common to all MDPs. The first example of this knowledge is that movement transitions are deterministic and yield no reward when the agent is traversing the grid. To make this precise, we define the set of states in the interior of the grid:

$$\mathcal{S}_{\text{interior}} := \left\{ s \mid |s_0| < \frac{N_{\text{grid}} - 1}{2}, |s_1| < \frac{N_{\text{grid}} - 1}{2}, \{s_i\}_{i \geq 2} = 0 \right\},$$

that is their location is not adjacent to the grid's boundary. All other values $s_2 : s_{1+N_{\text{victims}}+N_{\text{hazards}}}$ are set to 0. We define the set of movement actions to be $\mathcal{A}_{\text{movement}} := \{up, down, left, right\}$. Taking an action $a \in \mathcal{A}_{\text{movement}}$ when in state $s \in \mathcal{S}_{\text{interior}}$ always moves the agent in the direction of the action selected without changing the other states and receives a reward of 0, that is:

$$P_S(s \in \mathcal{S}_{\text{interior}}, a = up) = \delta(s'_1 = s_1 + 1, s'_{\neq 1} = s_{\neq 1}),$$
$$P_S(s \in \mathcal{S}_{\text{interior}}, a = down) = \delta(s'_1 = s_1 - 1, s'_{\neq 1} = s_{\neq 1}),$$
$$P_S(s \in \mathcal{S}_{\text{interior}}, a = right) = \delta(s'_0 = s_0 + 1, s'_{\neq 0} = s_{\neq 0}),$$
$$P_S(s \in \mathcal{S}_{\text{interior}}, a = left) = \delta(s'_0 = s_0 - 1, s'_{\neq 0} = s_{\neq 0}),$$
$$P_R(s \in \mathcal{S}_{\text{interior}}, a \in \mathcal{A}_{\text{movement}}) = \delta(r = 0),$$

allowing us to sample from $\mathcal{A}_{\text{movement}} \times \mathcal{S}_{\text{interior}} \subset \mathcal{D}_{\text{prior}}$ and apply the above transformation.

In addition to the deterministic transitions, we can also include prior reward information. Firstly we define the boundary states where the agent is adjacent to the edge of the grid

$$\mathcal{S}_{\text{boundary}} := \left\{ s \mid |s_0| = \frac{N_{\text{grid}} - 1}{2}, \{s_i\}_{i \geq 2} = 0 \right\} \cup \left\{ s \mid |s_1| = \frac{N_{\text{grid}} - 1}{2}, \{s_i\}_{i \geq 2} = 0 \right\}$$

We note that again all non-locations states $s_2 : s_{1+N_{\text{victims}}+N_{\text{hazards}}}$ are set to 0 because other values are specific to each MDP. As agents and hazards are initialised uniformly in the squares adjacent to the grid, if an agent is in $\mathcal{S}_{\text{boundary}}$ and takes an action to move out of the grid (i.e. open a door), then the expected reward will be:

$$r_{\text{prior}} := \frac{N_{\text{victims}}}{4 \times N_{\text{grid}}} \cdot r_{\text{victim}} + \frac{N_{\text{hazards}}}{4 \times N_{\text{grid}}} \cdot r_{\text{hazard}}.$$

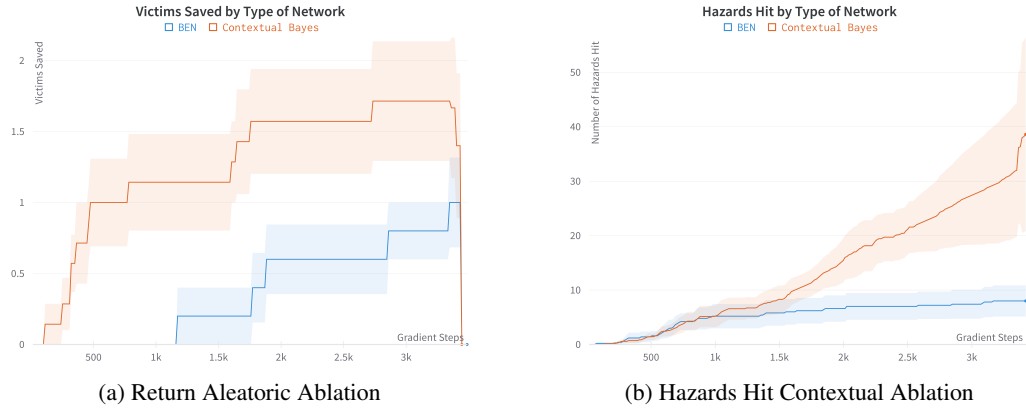

(a) Return Aleatoric Ablation        (b) Hazards Hit Contextual Ablation

Figure 10: Contextual vs BEN Ablation

**Listening Information**    Key to solving the search and rescue environment is learning to listen before acting.

### E.6   SEARCH AND RESCUE IMPLEMENTATION DETAILS

The epistemic network consists of two layers of ActNorm, a Masked Autoregressive Flow with two blocks and a LU linear decomposition and permutation as in Dinh et al. (2017). The base distribution is a unit Gaussian. This takes the number of parameters in the Aleatoric Network is a projection to 2d space from 1d, an Inverse Autoregressive Flow, a LU Linear decomposition and Permutation, a projection back to 1d with the Slice Flow, and an Abs Flow. The base distribution is a standard 1d Gaussian. The AbsFlow consists of 6 applications of the conditioner network, (K=6), and two layers. We vary the number of steps for the MSBBE minimisation with a learning rate of 1e-4 using ADAM for the stochastic gradient descent and use a separate ADAM optimiser, with a learning rate of 1e-4 for the Epistemic Network training on the ELBO. For the for Q-function approximator, we use a fully connected linear layer with ReLU activations, a gated recurrent unit and a final fully connected linear layer with ReLU activations. All hidden dimensions are 64. The dimension of the hidden state $\hat{h}_0$ is 64. The input size is the state space size 14 (4 number of victims + 8 number of hazards + 1 + 1 for x and y dims) The input dimension is state space size + 1 for reward + 1 for action = 16. The network output is 5 dimensional to reflect the five possible actions the agent can take. The input to the conditioner network is number of aleatoric parameters+ hidden dim size + 1 for q value, and the hidden layers and output size are the number of aleatoric parameters. Only a subset of these aleatoric parameters are used as needed in each layer and the rest are dropped.

### E.7   ABLATIONS

We carry out the following ablations in the zero-shot setting for the search and rescue environment, averaged over 7 seeds in this zero-shot test and plot the sample standard errors:

**Contextual Approaches**    We repeat the ablation carried out in the Tiger Problem for this new domain, demonstrating the existing approaches that learn a contextual optimal policy (i.e. state of the art model-free approaches such as BBAC (Fellows et al., 2021) and BootDQN+Prior (Osband et al., 2019)) cannot succeed in this challenging setting. This corresponds to using a function approximator with no capacity to represent history. BEN provides a clear improvement over these existing methods in terms of cumulative return in Fig. 2. To understand why, for each approach we plot the number of victims rescued in Fig. 10a and hazards hit in Fig. 10b. Although both approaches save a similar number of victims, the contextual approach hits an order of 10 times more hazards than BEN.These results demonstrate that contextual approaches struggle to solve this problem whereas BEN is slightly more conservative, yet does not hit nearly as many hazards, as we would expect given the disproportionally greater negative reward for hitting a hazard than rescuing the victim in our environment.

**Capacity for Representing Aleatoric Uncertainty**    We now investigate how reducing increasing the capacity of our aleatoric network affects the performance in this domain. We increase the number of aleatoric flow layers from 1 to 4 and plot the returns in Fig. 12a and the number of victims rescued in Fig. 12b. We see that for this environment, 2 flow layers yields the best returns. As the number

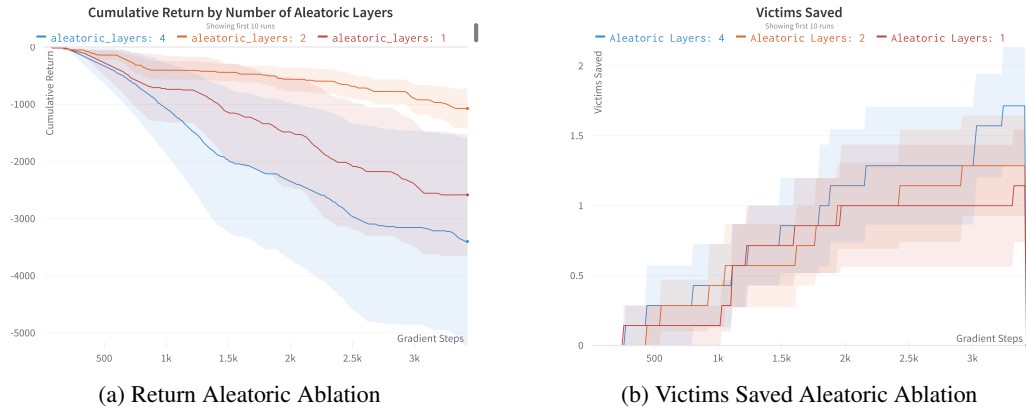

(a) Return Aleatoric Ablation                    (b) Victims Saved Aleatoric Ablation

Figure 11: Aleatoric Network Ablation

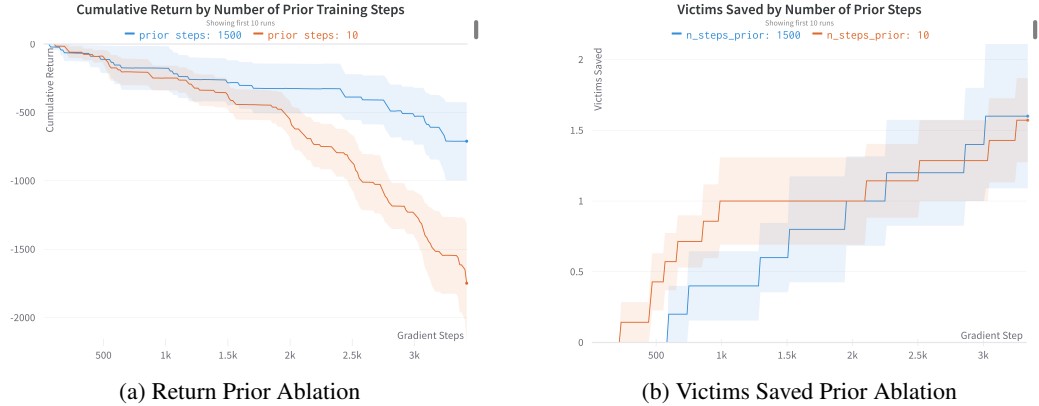

(a) Return Prior Ablation                         (b) Victims Saved Prior Ablation

Figure 12: Aleatoric Network Ablation

of aleatoric flow layers is determines the hypothesis space, our results provide evidence that there exists a trade-off between specifying a rich enough hypothesis space and a hypothesis space that is too general for the problem setting. For 4 layers, the hypothesis space is too general to learn how to behave optimally given the number of minimisation steps whereas for 1 layer, the agent cannot represent aleatoric uncertainty sufficiently to learn a policy that is useful for the environment. This ablation also supports our central claim that aleatoric uncertainty cannot be neglected in model-free Bayesian approaches.

**Incorporation of Prior Knowledge**     Finally, we investigate how our prior training regime affects the performance of BEN, varying the number of prior gradient training steps according to Algorithm 2. Results are plotted in Appendix E.7. A key motivation for taking a Bayesian approach to RL is the ability to formally exploit prior knowledge. We use this ablation to demonstrate how knowledge provided by simple simulations can be incorporated into BEN's pre-training regime. As we decrease the number of prior pretraining MSBBE minimisation steps, we see that performance degrades in the zero-shot settling as expected. Moreover, this ablation shows that a relatively few number of pre-training steps are needed to achieve impressive performance once the agent is deployed in an unknown MDP, supporting our central claim that BEN is computationally efficient.

