# OpenReview forum: "Bayesian Exploration Networks"
_ICLR.cc/2024/Conference — ICLR 2024 Conference Withdrawn Submission_

### Official Review · Reviewer_Z7MF · 2023-10-29

**Soundness:** 1 poor
**Presentation:** 1 poor
**Contribution:** 2 fair
**Rating:** 3
**Confidence:** 4

**Summary:**

This paper proposes a novel model-free method for learning Bayes-optimal policies. It is found through theoretical analysis that existing model-free methods fails to address the epistemic uncertainty in the future or optimizes over a set of contextual policies. The proposed Bayesian exploration network addresses these gaps by simultaneously modeling both epistemic and aleatoric uncertainties through the use of normalizing flows. The method has potential to learn the Bayes-optimal in the limit of complete optimization.

**Strengths:**

The paper introduces a novel method that models epistemic and aleatoric uncertainties using normalizing flows. This could be a valuable contribution to the field of reinforcement learning.

**Weaknesses:**

1.  **Mathematical Rigor:** Although the paper provides extensive mathematical analysis, there's a discernible lack of mathematical rigor.

*   _Theorem 1:_ The theorem hinges on Lemma 1. However, Lemma 1's proof is questionable as its final equality doesn't hold true. The MDP parameter, $\phi$, doesn't impact future decisions made by the contextual policy, which depend solely on history.
*   _Theorem 2:_ The theorem states that the posterior of a mis-specified deterministic model concentrate on the maximum likelihood estimation (MLE). However, it is obvious that given few data, there could be multiple models fit the data perfectly. All of them are MLE but not all MLE has non-zero posterior probability. Besides, if no model can fit the data perfectly, then the likelihood of data under any model are always $0$, implying that marginal likelihood of the data is also $0$. In this case, the posterior probability is the probability of the model conditioned on an impossible event, which is undefined.
*   _Theorem 3:_ The third theorem is based on the presumption that $b_t$ is a sufficient statistic. However, this is not the case if no further assumptions are made. We can easily construct examples where the pair $(s_{t+1},r_t)$ cannot be differentiated by just looking at $b_t=r_t+\gamma \sup_{a’}Q^*(h_{h+1},a’)$.

2.  **Exploration-Exploitation Dilemma:** The goal of this paper is to learn a Bayes-optimal policy that strikes a balance between exploration and exploitation. But the process of learning this policy through environmental interactions has its own exploration-exploitation quandary, which isn't adequately tackled.

3.  **Structure and Presentation:** The paper's layout is unwieldy. While the bulk of main text reiterates well-established and well-recognized results, critical details about the new method are relegated to the appendix. Consequently, understanding the new approach demands a thorough read of the appendix.

4.  **Related Work:** The paper overlooks model-free meta-reinforcement learning methods, such as those in [1], which align closely with its theme. Additionally, RL methods for POMDPs, e.g., [2], are not mentioned. It is not clear how the proposed method performs compared to methods that directly solve the history-MDP.


**Minor Points:**

*   Notations are introduced without prior definitions, such as $P_Q$ in Definition 1 and $P_{al}$ in page 19.
*   Definition 1: Replace "... over and a model …" with "... over a model …"
*   Definition 1 & Last paragraph of Sec 3: The term $P_Q$ is ambiguous. Typically, a Q-function wouldn't yield a distribution.
*   First paragraph in Sec 4.1: Change $\pi(\cdot,\theta)$ to $\pi(\cdot,\phi)$
*   Theorem 3: The phrase "be a measurable mapping $\mathcal{S}\times\mathbb{R}\to\mathbb{R}$" is initially perplexing and warrants elaboration.


[1] Beck, J., Vuorio, R., Liu, E.Z., Xiong, Z., Zintgraf, L., Finn, C. and Whiteson, S., 2023. A survey of meta-reinforcement learning. arXiv preprint arXiv:2301.08028.
[2] Hausknecht, M. and Stone, P., 2015, September. Deep recurrent q-learning for partially observable mdps. In 2015 aaai fall symposium series.

**Questions:**

How come the variational posterior isn't history-dependent?

---

> ### Author Response · Authors · 2023-11-17
> **Response to Reviewer Z7MF - Lemma 1**
>
> We thank the reviewer for their comments and detailed review and thank them for pointing out typos and points of clarification. We respond to detailed feedback regarding the validity of our mathematical results over two separate comments:
>
> The reviewer questions the validity of the final line and states: `The MDP parameter $\phi$ doesn't impact future decisions made by the contextual policy, which depend solely on history.'
>
> We're not sure what the reviewer means by this. If they are claiming our formulation is myopic, this can be proved as not the case by constructing the equivalent formulation using the myopic transition distributions. In the myopic formulation, the current posterior at timestep $t$ is used to marginalise across all future timesteps $t'>t$:
> $$
> p(r_{t'}\vert s_{t'},a_{t'},h_t)=\int p(r_{t'}\vert s_{t'},a_{t'} ,\phi) dP_{\Phi}(\phi\vert h_t),$$
>
> $$p(s_{t'+1}\vert s_{t'},a_{t'},h_t)=\int p(s_{t'+1}\vert s_{t'},a_{t'} ,\phi) dP_{\Phi}(\phi\vert h_t).$$
> Substituting gives the density over trajectories:
> \begin{align}
> 	p(\tau_{t:\infty}\vert h_{t},a_{t})=\prod_{i=t}^\infty p(s_{i+1}\vert s_{i},a_i,h_t)p(r_i\vert s_{i},a_i,h_t) \pi(a_{i+1}\vert s_{i+1},h_t),
> \end{align}
> which clearly cannot be factored to give the result in Lemma 1.
>
> Maybe it would be easier to explain our result by taking the Bayesian RL objective as a starting point. It is well established that the Bayesian RL objective can be written as an outer prior marginalisation over an inner expectation:
> \begin{align}
> 	J^\pi_\textrm{Bayes}=\int \int \sum_{i=0}^\infty \gamma^i r_i dP_{\infty}^\pi(\tau_\infty\vert \phi) dP_\Phi(\phi).
> \end{align}
> where:
> \begin{align}
> p_{\infty}^\pi(\tau_\infty\vert \phi)=p_0(s_0)\prod_{i=0}^\infty\pi(a_i\vert  h_i) p(s_{i+1}\vert s_i,a_i,\phi) p(r_i\vert s_i,a_i,\phi).
> \end{align}
> In this form, the BAMDP can be solved by sampling MDPs from the prior, and training a history dependent policy / Q function on them (this is precisely what naively applying a POMDP solver like RL$^2$ [Duan et al., 17] does as we mention in the paper). Just because the objective can be written in this form does not mean that $\phi$ does not impact future decisions.
>
> If we now limit the space of policies to contextual policies, i.e. policies that can be factored as $\pi(a_t\vert h_t) =\int \pi(a_t\vert s_t,\phi) dP_\Phi(\phi\vert h_t )$, elementary laws of probability show that under the expectation,
> \begin{align}
> 	J^\pi_\textrm{Contextual}=\int \int \sum_{i=0}^\infty \gamma^i r_i dP_{\infty}^\pi(\tau_\infty\vert \phi) dP_\Phi(\phi),
> \end{align}
> where
> \begin{align}
> 	p_{\infty}^\pi(\tau_\infty\vert \phi)=p_0(s_0)\prod_{i=0}^\infty\pi(a_i\vert  s_i,\phi) p(s_{i+1}\vert s_i,a_i,\phi) p(r_i\vert s_i,a_i,\phi).
> \end{align}
> If a proof of this is needed, consider:
> \begin{align}
> 	\int \int f(\tau_2) p^\pi_2(\tau_2\vert \phi)  p(\phi) d\phi d\tau_2=\int f(\tau_2) \int  p^\pi_2(\tau_2\vert \phi)  p(\phi) d\phi d\tau_2.
> \end{align}
> Now:
> $$
> 	\int  p^\pi_2(\tau_2\vert \phi)  p(\phi) d\phi =\int p_0(s_0)\pi(a_0\vert  s_0)\pi(a_1\vert h_1)   p(s_2,r_1\vert s_1,a_1,\phi) p(s_1,r_0\vert s_0,a_0,\phi) p(\phi) d\phi,$$
> $$
> 	=\pi(a_0\vert  s_0)\pi(a_1\vert h_1)   \int p(s_2,r_1, s_1,r_0,s_0\vert a_1,a_0,\phi) p(\phi) d\phi,
> $$
> $$
> 		=\pi(a_1\vert h_1)    \pi(a_0\vert  s_0) p(s_2,r_1, s_1,r_0,s_0\vert a_1,a_0),
> $$
> $$
> 	=\int  \pi(a_1\vert s_1,
> \phi) p(\phi\vert h_1) d\phi  \pi(a_0\vert  s_0)p(s_2,r_1,s_1,r_0,s_0\vert a_0),
> $$
> $$
> 	=\int  \pi(a_1\vert s_1,
> \phi) \frac{p(h_1\vert \phi)  p(\phi)}{p(h_1)}d\phi p(h_1),
> $$
> $$
> 	=\int  \pi(a_1\vert s_1,
> \phi) p(h_1\vert \phi)  p(\phi)d\phi ,
> $$
> $$
> 	=\int p_0(s_0)\pi(a_0\vert  s_0,\phi) p(s_1,r_0\vert s_0,a_0,\phi) \pi(a_1\vert  s_1,\phi)p(s_2,r_1\vert s_1,a_1,\phi)  p(\phi)d\phi.
> $$
>  Applying the same proof but conditioning on $h_t,a_t$ recovers our result from Lemma 1. We are more than happy to include a full step by step derivation if the reviewer still feels it is necessary.

---

> ### Author Response · Authors · 2023-11-17
> **Response to Reviewer Z7MF - Further Points**
>
> MLE Estimator:
>
>  We would like to emphasise that Theorem 2 is an example demonstrating the need to model aleatoric uncertainty. The issues that the reviewer takes can be resolved by assuming that the (empirical) projection exists and is unique:
>  \begin{align}
>  	\phi^\star_\textrm{MLE}(h_t)\in\arg\inf_{\phi\in\Phi}\left(\sum_{i=0}^{t-1} (b_i-B(h_i,a_i,\phi)^2\right).
>  \end{align}
>  This is an assumption that is commonly (and most typically implicitly [4,5,6]) made in works that analyse fitted/projection operator methods [1,2,3]. Again, we emphasise that the purpose of the theorem is to show why modelling aleatoric uncertainty is essential, as even under somewhat contrived assumptions, the best we can hope for is a policy that has no exploratory properties. We thank the reviewer for bringing this to our attention.
>
> Sufficiency of $b_t$:
>
> We would like to clarify what we mean here: our goal is to find an optimal policy and we propose that either $b_t$ or $r_{t}$ and $s_{t+1}$ is a sufficient statistic for solving the underlying RL problem (that is finding $\pi^\star$), not that $b_t$ is a sufficient statistic for  $r_{t}$ and $s_{t+1}$. This must hold or else model-free methods in general, which estimate $b_t$, would not be sufficient for learning optimal policies. We recognise that this is confusingly worded and thank the reviewer for bringing this to our attention. We will amend this and emphasise this key difference in an updated version.
>
>
> Related Work:
>
> We are aware of the review of meta RL cited by the reviewer, however we are not sure which methods they are referring to. We take care to emphasise that naively applying ANY POMDP solver to solve a BAMDP when taking a model-based approach, regardless whether the solver itself uses a model-free or a model-based approach, is not feasible for reasons of tractability. The only tractable methods that attempt this are based on VariBAD, which, as we explain, forfeit being Bayesian over part of their model. We took the time to carefully explain that model-free Bayesian approaches model uncertainty in the Bellman operator (see Definition 1).  None of the methods presented in the review do this. We emphasise that just because a POMDP solver uses model-free methods to solve the BAMDP does not mean that it is a model-free approach to Bayesian RL. Model-free Bayesian RL approaches characterise uncertainty in the variable $b_t$. We are happy to add the review to our list of citations.
>
> [1] Antos et al.Learning near-optimal policies with Bellman-residual minimization based fitted policy iteration and a single sample path. Machine Learning, 07.
>
> [2] Antos et al. Fitted Q-iteration in continuous action-space MDPs. NeurIPS 07
>
> [3] A. Antos et al. Value-iteration based fitted policy iteration: learning with a single trajectory. IEEE 07
>
> [4] Riedmiller, Neural Fitted Q Iteration - First Experiences with a Data Efficient Neural Reinforcement Learning Method,  ECML 05
>
> [5] Sutton et al. Fast Gradient-Descent Methods for Temporal-Difference Learning with Linear Function Approximation. ICML 09
>
> [6] Maei et al. Convergent Temporal-Difference Learning with Arbitrary Smooth Function Approximation. NeurIPS 09

---

> > ### Comment · Reviewer_Z7MF · 2023-11-18
> > **Response to rebuttal**
> >
> > Thanks for your response.
> >
> > On Lemma 1: It is said in the response that elementary laws of probability show that under the expectation,
> > $$P_\infty^\pi(\tau_\infty|\phi)=p_0(s_0)\prod_{i=0}^\infty\pi(a_i|h_i)p(s_{i+1}|s_i,a_i,\phi)p(r_i|s_i,a_i,\phi)$$
> > $$=p_0(s_0)\prod_{i=0}^\infty p(s_{i+1}|s_i,a_i,\phi)p(r_i|s_i,a_i,\phi)\int \pi(a_i|s_i,\phi)dP_\phi(\phi|h_i)$$
> > $$=p_0(s_0)\prod_{i=0}^\infty p(s_{i+1}|s_i,a_i,\phi)p(r_i|s_i,a_i,\phi)\pi(a_i|s_i,\phi),$$
> > which is, however, apparently incorrect. The issue arises because $\phi$ inside the integral is a variable over which you are integrating, while $\phi$ outside the integral is a fixed parameter. This mismatch means that $\phi$ inside and outside the integral are not the same entity and should not be treated as such.
> >
> > In the provided proof for the $t=2$ case, I found multiple errors. Here is my derivation for the last four equalities,
> > $$\pi(a_1|h_1)\pi(a_0|s_0)p(s_2,r_1,s_1,r_0,s_0|a_1,a_0)$$
> > $$=\int \pi(a_1|s_1,\phi)p(\phi|h_1)d\phi \pi(a_0|s_0)p(s_2,r_1,s_1,r_0,s_0|a_1,a_0)$$
> > $$=\int\pi(a_1|s_1,\phi)\frac{p(\phi)p(s_1,r_0,s_0,a_0|\phi)}{ p(s_1,r_0,s_0,a_0)}d\phi p(s_2,r_1,s_1,r_0,s_0, a_0|a_1)$$
> > $$=\int\pi(a_1|s_1,\phi)p(\phi)p(s_1,r_0,s_0,a_0|\phi) d\phi \frac{p(s_2,r_1,s_1,r_0,s_0, a_0|a_1)}{p(s_1,r_0,s_0,a_0)}.$$
> >
> > On Theorem 2: If the mean squared error is to be taken as the negative likelihood, then the posterior would not be degenerated as intended. If you insist in using Dirac delta as the likelihood function, then the posterior should be what I stated in the review. By the way, I believe this theorem is of no avail even if it may be fixed in some way. The assumption looks contrived, and the conclusion feels apparent. As I have suggested in the review, I believe contents like this should make way for critical details of the proposed method.
> >
> > On Theorem 3: I am not convinced that $b_t$ is sufficient in inferring $\phi$, which is exploited in Equation (10). Can you provide a formal proof?

---

> ### Author Response · Authors · 2023-11-20
> **Lemma 1**
>
> We think the confusion may be coming from the fact Lemma 1 holds only because we are restricting ourselves to contextual policies. For simplicity, consider the space of contextual optimal policies, as this is what Theorem 1 analyses. As we write in our paper, a contextual optimal policy is defined as:
> \begin{align}
> 	\pi^\star(\cdot,\phi)\in \arg \max_{\pi\in\Pi_\Phi} J^\pi(\phi).
> \end{align}
> Given an MDP indexed by $\phi$, we can always find a contextual optimal policy by taking the optimal action $a\in\arg\max_{a'} Q^\star(s,a,
> \phi)$, where:
> \begin{align}
> 	Q^\star(s,a,\phi)=E_{\tau_\infty \sim P_\infty^\star (s,a,\phi)} \left[\sum_{i=0}^\infty \gamma^i r_i \right].
> \end{align}
> and:
> \begin{align}
> 	p_\infty^\star (\tau_\infty\vert s,a,\phi)=\prod_{i=0}^\infty p(r_i\vert s_i,a_i,\phi)p(s_{i+1}\vert s_i,a_i,\phi) \pi^\star(a_{i+1}\vert s_{i+1},\phi).
> \end{align}
> Finding $Q^\star(s,a,\phi)$ can be achieved by solving the contextual optimal Bellman equation 	$B[Q^\star](s,a,
> \phi)=Q^\star(s,a,
> \phi)$ where:
> \begin{align}
> 	B[Q^\star](s,a,
> 	\phi)=E_{s',r\sim P_{R,S}(s,a,\phi)}[r+\gamma\sup_{a'} Q^\star(s',a',\phi)].
> \end{align}
>  We can thus (in principle) learn a set of contextual optimal policies by learning $Q^\star(s,a,
> \phi)$ and $\pi^\star(\cdot,\phi)$ for all $\phi\in \Phi$ by solving the afformentioned optimal contextual Bellman equation. Existing model-free BRL methods then characterise the uncertainty in $	B[Q^\star](s,a,
> \phi)$ by first sampling $s',r\sim P_{R,S}(s,a,\phi^\star)$ from the environment and then applying the transformation $b=r+\gamma\sup_{a'} Q^\star(s',a',\phi)$. A posterior over $\phi$ can be inferred $p(\phi\vert h_t)$, which is used to marginalise across all optimal Bellman operators $B[Q^\star](h_t,
> a_t)=E_{\phi \sim {P_\Phi}( h_t) } [ B [ Q^\star ] ( s_t,a_t,
> \phi) ] $
>
> In practice, existing methods partially solve the corresponding contextual Bellman equation online $B[Q^\star(h_t,a_t)=Q^\star(h_t,a_t)$ after inferring each $p(\phi\vert h_t)$ using variational inference and employing posterior sampling to yield a tractable algorithm. Regardless, the goal of these approaches is still to find:
> \begin{align}
> 	B[Q^\star](h_t,
> a_t)=	E_{\phi\sim P_\Phi( h_t) }[B[Q^\star](s_t,a_t,
> \phi)]=E_{\phi\sim P_\Phi( h_t)}[Q^\star(s_t,a_t,\phi)].
> \end{align}
> Now:
> \begin{align}
> 	E_{\phi\sim P_\Phi( h_t) }[Q^\star(s_t,a_t,\phi)]=	E_{\phi\sim P_\Phi( h_t) }\bigg[E_{\tau_{t:\infty} \sim P_{t:\infty}^\star ( s_t,a_t,\phi)} \bigg[\sum_{i=t}^\infty \gamma^i r_i \bigg]\bigg]
> 	=	E_{\tau_\infty \sim P_{t:\infty}^\star (h_t,a_t)} \bigg[\sum_{i=0}^\infty \gamma^i r_i \bigg],
> \end{align}
> where:
>  \begin{align}
> 	p_{t:\infty}^\star (\tau_{t:\infty}\vert h_t,a_t)= \int \prod_{i=0}^\infty p(r_i\vert s_i,a_i,\phi)p(s_{i+1}\vert s_i,a_i,\phi) \pi^\star(a_{i+1}\vert s_{i+1},\phi) dP_\Phi(h_t).
> \end{align}
> and so the proof of Lemma 1 holds, but only because we are limiting ourselves to the space of contextual optimal policies.

---

> > ### Comment · Reviewer_Z7MF · 2023-11-20
> >
> > This new definition of contextual policies clearly diverges from the definition in the paper, which writes that $\Pi^*_{Contextual}\coloneq \arg\sup_{\pi\in\Pi_{Contextual}}J_{Bayes}^\pi$.
> > Under the new definition, contextual policies are basically QMDP policies introduced in [1], which makes Theorem 1 a tautology.
> >
> > By the way, I notice that, in Section 4.2, the presented myopic policies are referred to as QMDP, which is incorrect. The so-called myopic policies are learned assuming future epistemic uncertainty remains unchanged, while QMDP assumes the epistemic uncertainty disappears after one step.
> >
> > [1] Littman, Michael L., Anthony R. Cassandra, and Leslie Pack Kaelbling. 1995. “Learning Policies for Partially Observable Environments: Scaling Up.” In Machine Learning Proceedings 1995, 362–70.

---

> > > ### Author Response · Authors · 2023-11-21
> > >
> > > The example restricting to contextual-optimal policies was a demonstration of the proof of Lemma 1, which is what you took issue with.
> > >
> > >
> > > Re-reading the reviewer's concerns, we think we have identified a key point of confusion which is why they believe our proof is wrong. Unlike a general POMDP, in a BAMDP the parameter transition is deterministic [1]  because the underlying MDP is not changing every timestep, that is:
> > > \begin{align}
> > > 	p(\phi_{t+1}\vert  \phi_t,h_t) =\delta(\phi_{t+1}=\phi_t).
> > > \end{align}
> > >
> > > We will prove Lemma 1 by going right back to the definition of the BAMDP and deriving the Bayesian Q-function.
> > > For notational simplicity, let $x_t\coloneqq (a_t,r_t,s_{t+1})$ and hence $h_t=(s_0,x_0,x_1, ...x_{t-1})$. Under this notation and our assumption of restriction to contextual policies:
> > > \begin{align}
> > > 	p(x_t\vert s_t, \phi_t)=\pi(a_t\vert s_t,\phi_t)p(r_t\vert s_t,a_t,\phi_t)p(s_{t+1}\vert s_t,a_t,\phi_t)
> > > \end{align}
> > >
> > >
> > > We can write the belief update as:
> > > \begin{align}
> > > 	p(h_{t+1}\vert h_t)=p(h_t,x_t\vert h_t)=p(x_t\vert h_t)
> > > \end{align}
> > > where:
> > > \begin{align}
> > > 	p(x_t\vert h_t)=\int 	p(x_t\vert s_t, \phi_t)p(\phi_t\vert h_t)d\phi_t
> > > \end{align}
> > > \begin{align}
> > > =\int 	p(x_t\vert s_t, \phi_t)\int p(\phi_t,\phi_{t-1}\vert h_t) d\phi_{t-1}d\phi_t
> > > \end{align}
> > > \begin{align}
> > > 	=\int \int 	p(x_t\vert s_t, \phi_t) p(\phi_t,\phi_{t-1}\vert h_t)d\phi_t d\phi_{t-1}
> > > \end{align}
> > > \begin{align}
> > > 	=\int \int p(x_t\vert s_t, \phi_t) p(\phi_t\vert \phi_{t-1}, h_t) p(\phi_{t-1}\vert h_t)d\phi_t d\phi_{t-1}
> > > \end{align}
> > > \begin{align}
> > > 	=\int \int p(x_t\vert s_t,\phi_t) \delta(\phi_t= \phi_{t-1}) p(\phi_{t-1}\vert h_t)d\phi_t d\phi_{t-1}
> > > \end{align}
> > > \begin{align}
> > > 	= \int p(x_t\vert s_t,\phi_{t-1 }) p(\phi_{t-1}\vert h_t)d\phi_{t-1}
> > > \end{align}
> > > Recursively applying yields:
> > > \begin{align}
> > > 	p(x_t\vert h_t)=\int p(x_t\vert s_t, \phi_0) p(\phi_0\vert h_t)d\phi_0
> > > \end{align}
> > > Now,
> > > \begin{align}
> > > 	p(\tau_{t:t'}\vert h_t,a_t)=p(x_{t'},h_{t'}\vert h_t,a_t)=p(x_{t'}\vert h_{t'})p(h_{t'}\vert h_t,a_t)
> > > \end{align}
> > > Applying the above result and Bayes' rule yields:
> > > \begin{align}
> > > p(\tau_{t:t'}\vert h_t,a_t)=\int p(x_{t'}\vert s_{t'},\phi_0) p(\phi_0\vert h_{t'})d\phi_0 p(h_{t'}\vert h_t,a_t)
> > > \end{align}
> > > \begin{align}
> > > =\int p(x_{t'}\vert s_{t'},\phi_0) \frac{p(h_{t'}\vert h_t,a_t,\phi_0)p(
> > > \phi_0\vert h_t,a_t)}{p(h_{t'}\vert h_t,a_t)}d\phi_0 p(h_{t'}\vert h_t,a_t)
> > > \end{align}
> > > \begin{align}
> > > 	=\int p(x_{t'}\vert s_{t'},\phi_0) p(h_{t'}\vert h_t,a_t,\phi_0)p(
> > > 		\phi_0\vert h_t,a_t)d\phi_0
> > > \end{align}
> > > \begin{align}
> > > 	=\int p(h_{t'}\vert h_t,\phi_0)p(
> > > 	\phi_0\vert h_t,a_t)d\phi_0
> > > \end{align}
> > > \begin{align}
> > > 	=\int p(\tau_{t:t'}\vert s_t,a_t,\phi_0)p(
> > > 	\phi_0\vert h_t)d\phi_0
> > > \end{align}
> > > Substituting into the definition of the Bayesian $Q$-function recovers our result:
> > > \begin{align}
> > > 	Q^\pi(h_t,a_t)=\int\int \sum_{i=t}^\infty r_i\gamma^i  p(\tau_{t:\infty}\vert s_t,a_t,\phi_0)p(
> > > 	\phi_0\vert h_t)d\phi_0 d\tau_{t:\infty}
> > > \end{align}
> > > \begin{align}
> > > =\int\int  \sum_{i=t}^\infty r_i\gamma^i  p(\tau_{t:\infty}\vert s_t,a_t,\phi_0)  d\tau_{t:\infty}p(
> > > 	\phi_0\vert h_t)d\phi_0
> > > \end{align}
> > > \begin{align}
> > > =\int Q^\pi (s_t,a_t,\phi_0)p(
> > > 	\phi_0\vert h_t)d\phi_0
> > > \end{align}
> > > as required. The proof is not tautological, it states the important result: approaches that model uncertainty in an optimal Bellman operator or optimal Q-function using existing methods do propagate uncertainty like Bayes-optimal policies but they are restricted to planning using the set of contextual policies. This result has never been proved before, as it has not been established whether existing model-free approaches are Bayes-optimal or not.
> > >
> > >
> > > [1] Ghavamzadeh et al, Bayesian Reinforcement Learning: A Survey, Chapter 4

---

### Official Review · Reviewer_Hvyz · 2023-11-03

**Soundness:** 3 good
**Presentation:** 2 fair
**Contribution:** 3 good
**Rating:** 6
**Confidence:** 2

**Summary:**

This paper proposes a model-free Bayesian RL method.

As opposed to maintaining and using a posterior over the MDP (transition/reward) model, this approach attempts to directly learn the (history-based) Q-function that is optimized with a Bayesian Bellman update.
While this update is defined as an expectation of the posterior over the MDP, the method avoids computing this posterior by instead learning the posterior over the Bellman update as a sufficient statistic.

This posterior is rather complex and, instead, they learn an approximation through variational inference with normalizing flow networks.
Now given this approximate posterior over the Bellman update, a typical RNN is used to learn the Bayesian Q-function.

**Strengths:**

Bayesian RL is an important line of work that provides a solution to the exploration but suffers from computational complexity.
Any progress on this front, as a result, should be relevant to a significant portion of ICLR's community.

Additionally, as described in the paper, while model-based methods have proliferated, model-free approaches have seen less attention.
Hence, this work is an important contribution.

An important aspect of this is the rigorous theory to support the rather novel perspective taken in the paper which is well supported (in the appendix).
Lastly, it is interesting to see that despite it being a model-free method, model-based prior knowledge is still an important aspect (which is positive because generally, those priors are much more practical to define).

**Weaknesses:**

Two key rooms for improvement are the clarity (presentation) and experimental section.

First, while the theoretical support in the appendix is certainly substantial, I found it rather difficult to follow key parts of the method description.
I believe this is, first, because the (general/theoretical) learning objective and its concrete (practical, normalizing flow network approximation) implementation are presented simultaneously.
Potential more important is the fact that the method description did not start until page 7 with background plus related work taking up much of the space which is a nice addition, but in this case in my opinion wrong priorities.

In terms of evaluation, the experiments were relatively bare (in terms of baselines an domains), limited in scope, and assumed a lot of prior knowledge.
For instance, the complete prior and posterior that a model-based approach would have used in the tiger problem is a distribution over two elements (the tiger is behind door 1 or door 2, with a uniform prior).
Furthermore, all the experiments are one-shot tasks.
Making it hard to estimate how good the method would do with less prior knowledge and in a more typical reinforcement learning setting.

As a result, in general, I found the paper more difficult to comprehend (than I believe necessary), and overall the empirical evaluation was lacking.
I do not know whether this is because some scaling difficulties stop the proposed solution from being tested on problems with less prior knowledge.

**Questions:**

N/A

---

> ### Author Response · Authors · 2023-11-17
> **Response to Reviewer Hvyz**
>
> We thank the reviewer for their feedback, especially regarding the structure of the paper. Due to the novelty of our approach, we have had often contractionary feedback about how to best present our work. We do really like the suggestion to shorten the background material and splitting up the theoretical and implementation details and will re-structure the paper according to this proposal. With regards to experiments, not all of our implementations are one-shot tasks. As discussed in Section 6 we evaluate BEN in the episodic setting for the search and rescue task too. We focused on the one-shot setting for our ablations as this more closely mimics what a rescue robot would have to achieve in real life (including being given training knowledge in the form of simulated related situations) and it highlights the important failures of existing model-free approaches to BRL.

---

> > ### Comment · Reviewer_Hvyz · 2023-11-20
> >
> > Thank you for your response and looking forward to the continuation of the discussion with the reviewer below!

---

### Official Review · Reviewer_MKkw · 2023-11-06

**Soundness:** 4 excellent
**Presentation:** 4 excellent
**Contribution:** 3 good
**Rating:** 8
**Confidence:** 3

**Summary:**

The authors propose a new model-free framework, Bayesian exploration networks, for performing Bayesian reinforcement learning. The first contribution is to show the shortcomings of existing model-free BRL methods. They show that these existing approaches don't learn Bayes-optimal policies because they don't properly propagate uncertainty through the MDP and only solve an approximation to the true Bayesian objective. Their second contribution is to propose a solution for these issue in the form of Bayesian exploration networks. The BEN framework simplifies the objective by using a Q-function approximator to reduce the dimensionality of the input, which is then passed to a Bayesian network. This results in a fewer number of parameters over which inference must be performed. Beyond theoretical results, they demonstrate the practical performance of BENs in a search and rescue problem.

**Strengths:**

- Clear communication. The authors present prior work and their own work in a clear and concise manner. The logical flow of the paper is very nice.
- The authors are very clear about the shortcomings of prior model-free BRL methods and how exactly their proposed approach addresses these shortcomings.
- The structure of BENs is not overly complicated. They use well-known building blocks, such as Q-function approximating functions and normalizing flows, to address the need to model uncertainty in certain parts of the method.

**Weaknesses:**

- While the authors do a nice job of reviewing prior literature, the magnitude of the contribution presented here is not clear. I am inclined to say that the importance of the authors' contributions is relatively low, although they are novel. The theoretical results showing the shortcomings of other model-free BRL approaches is arguably their most important contribution, but it's not clear that that's a sufficient contribution in isolation. I view their formulation of BENs as less impactful.
- The solution to circumventing costly nested optimization in Algorithm 1 is questionable. I would want to see more results that this is not harmful.
- Lack of comparison to methods. I would like to see further empirical evaluation of their approach, both in terms of environments tested and methods compared.

**Questions:**

In which practical situations would you genuinely seek to avoid existing model-free BRL methods? Do the theoretical shortcomings of existing approaches translate into material empirical shortcomings in common situations?

---

> ### Author Response · Authors · 2023-11-17
> **Response to Reviewer MKkw**
>
> We really thank the reviewer for their feedback. We are glad that they enjoyed reading the paper and appreciated their thoughtful criticisms. In response to their question about when we would genuinely seek to avoid existing model-free BRL methods, we believe that BEN offers a Pareto improvement on these methods and many can be recovered by using a linear flow. We believe that there might be slight benefits from a practitioner's perspective in using existing approaches: implementing BEN  requires expertise in normalising flows whereas existing methods are slightly simpler. For toy domains where exploration is difficult but the environment's dynamics are simple enough to avoid the pathological examples outlined in our work, for example in domains like Mountain Car, existing approaches will suffice.